# Application of Electrochemical Atomic Force Microscopy (EC-AFM) in the Corrosion Study of Metallic Materials

**DOI:** 10.3390/ma13030668

**Published:** 2020-02-03

**Authors:** Hanbing Chen, Zhenbo Qin, Meifeng He, Yichun Liu, Zhong Wu

**Affiliations:** 1School of Materials Science and Engineering, University of Shanghai for Science and Technology, Shanghai 200093, China; chb19940420@163.com; 2Key Laboratory of Advanced Ceramics and Machining Technology (Ministry of Education), Tianjin University, Tianjin 300072, China; qinzhb@tju.edu.cn; 3School of Materials Science and Engineering, Kunming University of Science and Technology, Kunming 650093, China; spsjtu@163.com

**Keywords:** EC-AFM, corrosion, metallic materials

## Abstract

Electrochemical atomic force microscopy (EC-AFM), a branch of a scanning probe microscopy (SPM), can image substrate topography with high resolution. Since its inception, it was extended to a wide range of research areas through continuous improvement. The presence of an electrolytic cell and a potentiostat makes it possible to observe the topographical changes of the sample surface in real time. EC-AFM is used in in situ corrosion research because the samples are not required to be electrically conductive. It is widely used in passive film properties, surface dissolution, early-stage corrosion initiation, inhibitor efficiency, and many other branches of corrosion science. This review provides the research progress of EC-AFM and summarizes the extensive applications and investigations using EC-AFM in corrosion science.

## 1. Introduction

As is known, all materials have a certain service life and will suffer various forms of direct or indirect damage during use. Although the material will be damaged in a variety of forms, corrosion is the most important and common form, which is a gradual process and cannot be restored. The problem of material corrosion occurs in various fields of the national economy, from daily life to industrial and agricultural production, as well as from advanced science and technology to the development of the national defense industry. Corrosion can lead to huge economic losses and even catastrophic accidents, which not only consume a large amount of resources and energy but also pollute the environment, resulting in a huge loss in the gross domestic product (GDP) [1]. Therefore, it is necessary to clarify the failure mechanism of materials in corrosive environments and take reasonable protective measures to achieve the purpose of preventing or controlling corrosion, to improve the life cycle of materials.

Corrosion is the irreparable damage or deterioration of materials caused by chemical, electrochemical, and physical effects of environmental media. In processes of metallic material corrosion, a chemical or electrochemical multiphase reaction occurs on the surface or interface of the metal, resulting in the conversion of the metal to an oxidized (ionic) state. Corrosion of metallic substance is started via the oxidation of metals.
M → M*^n^*^+^ + *n*e^−^,(1)
M + (x + y)H_2_O → MO_x_(OH)_y_ + (2x + y)H^+^ + (2x + y)e^−^.(2)

In order to maintain the neutral condition, a counter cathodic reaction of the electrolyte occurs [2].
2H_2_O + 2e^−^ → H_2_ + 2OH^−^,(3)
2H^+^ + 2e^−^ → H_2_,(4)
O_2_ + 2H_2_O + 4e^−^ → 4OH^−^,(5)
O_2_ + 4H^+^ + 4e^−^ → 2H_2_O,(6)
M*^n^*^+^ + *n*e^−^ → M.(7)

The basic process of metallic corrosion in an aqueous solution consists of the anodic dissolution of metals and the cathodic reduction of oxidants. The redox reactions (Equations (1)–(7)) involve the transfer of electrons and ions between the metal and the solution. According to the corrosion kinetics, the anodic oxidation current of the metal degradation is equal to the cathodic reduction current of the oxidant at the corrosion potential. When the metal electrode potential is more positive, the rates of cathodic reactions increase and the rates of anodic reactions decrease accordingly. Conversely, as the metal electrode potential becomes more negative, the effect on the reactions is opposite.

The development of corrosion science is inseparable from the advancement of research methods and instruments. Conventional electrochemical measurements can only obtain macroscopic electrochemical information on the surface of the material. In situ electrochemical scanning probe technology with high spatial resolution facilitated the development of corrosion science, elucidating the microstructure and dynamic properties of materials and interfaces at the molecular/atomic level.

The electrochemical atomic force microscope (EC-AFM) was developed in 1991 on the basis of the atomic force microscope (AFM) [3]. It is well known that AFM is a kind of scanning probe microscope (SPM) and an extension of the scanning tunneling microscope (STM) [4]. The STM was invented by Gerd Binnig and Heinrich Rohrer in the 1980s, and it was able to image individual atoms for the first time. In 1982, Binnig and co-workers indicated that vacuum tunneling with an externally controllable tunnel distance is feasible, even under the conditions of room temperature and non-ultra-high vacuum, which was the first step in the development of scanning tunneling microscopy [5]. The 7 × 7 reconstruction on Si(111) observed in real space with STM was considered the first scientific success [6]. G. Binnig and H. Rohrer went into detail about STM, and they were rewarded with the Nobel Prize in Physics for their work in the field of STM in 1986 [7,8]. At present, STM is still an important method in surface science. However, due to the operation principle of STM being based on the quantum tunneling effect, STM has a severe restriction that requires the samples to be conductive. STM can only directly observe the surface structure of conducting and semiconducting materials. In order to overcome the shortcomings of STM, the AFM was invented by Binnig, Quate, and Gerber in 1986 [4]. The AFM can image the surfaces of a flat sample through the weak interaction force (atomic force) between the probe (mounted to a cantilever) and the sample, which is measured by monitoring the deflection of the microcantilever. Therefore, the AFM is suitable for both conductive and non-conductive samples, and its application field is more extensive. In addition, the AFM is a high-resolution microscope with atomic-scale resolution, which developed into a powerful micro/nanometer-scale surface analysis technique [9,10]. On the basis of different capabilities, a large number of techniques were developed after the invention of AFM, such as Kelvin probe force microscopy (KPFM), magnetic force microscopy (MFM), scanning electrochemical microscopy/atomic force microscopy (SECM–AFM), electrostatic force microscopy (EFM), and EC-AFM. These technologies are not only imaging tools; they can also accurately and quantitatively measure local physical and chemical phenomena.

EC-AFM extends AFM technology to the field of electrochemistry for in situ studying of the solid–liquid interface. The main difference between EC-AFM and AFM in liquids is that the applied potential of the sample with respect to the reference electrode is controlled by a potentiostat [11]. Topographic changes of the sample are achieved by measuring the force between the tip and the substrate in a three-electrode electrochemical cell composed of the counter electrode, reference electrode, and working electrode (substrate). EC-AFM is applied to in situ electrodeposition [12,13], biological science [14], supercapacitors [15], batteries and electrodes [16], corrosion and protection, and so on. After the invention of EC-AFM, its electrolytic cell, imaging mode, and probe performance were continuously improved, which opens a new prospect for the understanding of the corrosion mechanism [17,18].

In this review, we illustrate the application of EC-AFM to metal corrosion with specific examples, including understanding the properties of passive films, surface dissolution, pitting corrosion, selective corrosion, intergranular corrosion, inhibitor efficiency, coating protection, and other branches of corrosion science. The main purpose of this review is to provide the present situation and trend of corrosion research on various metal materials by EC-AFM and outline future prospects for the technique.

## 2. Principle and Operation Modes of EC-AFM

The basic working principle of EC-AFM is similar to that of AFM, and it shares its key components. Primarily, a nanoscale sharp tip located at the end of the cantilever of the probe is used to feel the interaction force between the tip and the sample, a piezo scanner is used for controlling the movement of the tip or sample in the *x*-, *y*-, and *z*-directions, a feedback control system and feedback circuit are used to control the deflection of the cantilever, and a computer system is used to display the results and control the operational parameters [19,20]. In addition, EC-AFM requires an electrochemical cell that can accommodate the working electrode (WE), counter electrode (CE), and reference electrode (RE), as well as a potentiostat for normal potentiostat control in a three-electrode system. Generally, the electrochemical cell is made from chemical-resistant polycarbonate, which can be used with a wide variety of liquids. Eight-degree nose assemblies are recommended for imaging in liquid because the smaller angle considers the different angles at which the laser goes in and out of the fluid, compared to operation in air. The electrochemical cell typically contains retaining clips, an O-ring gasket, and a liquid cell plate. When assembled, the sample itself comprises the bottom of the liquid container. Therefore, the sample must be large enough for the O-ring to seat. The WE should be relatively small, and each point on the WE should be geometrically equivalent to the counter electrode CE, which ensures that the current and potential across the electrode are evenly distributed. Others are based on conventional electrochemical testing requirements. Figure 1 shows the diagram of a typical electrochemical AFM cell with a potentiostat. During the scanning, the sample to which the potential is applied is placed in the electrolytic cell as the working electrode and scanned with the tip mounted on the cantilever. The force between the tip and the sample is measured by monitoring the deflection of the cantilever. By plotting the deflection of the cantilever versus its position on the sample, a topographic image of the sample is obtained [21].

The operation mode of EC-AFM is also similar to that of AFM, and the instrument can be operated either in constant height mode or constant force mode. Based on the interaction force between the tip and the sample, it is generally divided into different operation modes in the imaging process, such as contact mode, non-contact mode, intermittent tapping (tapping mode), torsional resonance mode, and peak force tapping mode. The first is the contact mode, in which the tip of the probe is always in contact with the surface of the sample, and the force between the tip and the sample is a repulsive force. Since the silicon nitride cantilever is soft enough to be deflected and has a high resonance frequency to avoid vibration instability, a silicon nitride probe is often used in contact mode. The contact mode has the advantage of high resolution and high scan speed, but it may cause damage to the surface of the sample. Both the transverse shear force and the capillary force on the surface may have adverse effects [22]. The second type is the non-contact mode, in which the cantilever vibrates above the sample surface, and the distance between the tip and the sample is usually several nanometers [23]. The force between the tip and the sample is van der Waals attraction. In the non-contact mode, there is no damage to the surface of the sample and the lateral force is the smallest, but the resolution is low and the scanning speed is slow. In order to avoid being stuck to the water film on the sample surface, it is often used to scan hydrophobic surfaces and soft samples. However, the development and improvement of this device in recent years enabled non-contact mode to detect repulsion and obtain atomic resolution images. Miyata used a non-contact-mode AFM to observe the dissolution of calcite in water at high speed [24]. This may well be applicable to the observation of corrosion processes on metal surfaces. The third type is the intermittent tapping (tapping mode), in which the cantilever oscillates at a frequency close to its resonance frequency and the oscillation amplitude is monitored [25]. The cantilever oscillates near its resonant frequency above the surface of the sample, and the probe contacts the sample surface once at the bottom of the oscillation during each vibration period. The difference with the non-contact mode is that the amplitude (amplitude > 20 nm) of the cantilever is larger than that of the non-contact mode (amplitude < 10 nm). It not only reduces damage to the sample and reduces the effects of transverse force, but it also has a higher lateral resolution on most samples. The disadvantage is that the scanning speed is limited.

In addition, in order to further understand the properties of nanomaterials, a torsional resonance mode (TRmode) AFM was developed, which can measure both vertical and lateral force concurrently [26]. In the torsional resonance mode, the cantilever conducts torsional vibration with the long axis as the center and causes the tip to be in the vibration state. When the probe encounters transverse force on the surface, the system can detect the change in the cantilever torsion vibration and detect the fluctuation in the surface topography of sample [27]. It should be noted that the tapping mode applies a compressive force and the TR mode applies a torsional force; thus, the normal and shear performance is measured in the tapping mode and the TR mode, respectively [28]. Another newly developed operation mode is the peak force tapping mode, in which the tip is oscillated periodically in the vertical direction at a frequency well below cantilever resonance, tapping the sample until the maximum repulsion force reaches the setpoint in each tap [29]. The z piezo is driven with sinusoids rather than a triangular waveform in conventional force–distance (F–D) curves [30]. This mode directly controls the interaction between the tip and the sample, reducing the depth of deformation and the corresponding contact area, minimizing damage to the probe or sample [31,32]. Moreover, the force–distance interactions can be measured directly through peak force tapping mode.

Table 1 summarizes the operation mode of EC-AFM with the necessary parameters added. In the field of corrosion, contact mode and intermittent tapping (tapping mode) are mainly used.

## 3. Application of EC-AFM in Metal Corrosion and Corrosion Protection

### 3.1. The Micro-Area Corrosion

#### 3.1.1. The Corrosion Product Film

EC-AFM can characterize the real-time nanoscale topographic changes of a corrosion product film on a metal surface under different potentials in aqueous environments, which avoids the potential contamination and degradation of the corrosion product film during conventional ex situ characterization and analysis techniques.

A typical example of studying corrosion product films with EC-AFM was presented by Li and co-workers, who studied the surface passivation film and its substructure of carbon steel in a carbonate/bicarbonate solution by EC-AFM in contact mode [33]. They designed a solution container for the EC-AFM, which was large enough to accommodate the three-electrode system and avoid evaporation during testing. The topographic characterization of the sample surface was performed at the passivation potentials of −0.1 V_(SCE__—saturated calomel electrode)_, 0.5 V_(SCE)_ and 0.7 V_(SCE)_. The results showed that, at −0.1 V_(SCE)_ and 0.5 V_(SCE)_, as the passivation time increased, the nanoscale features, i.e., the scale-like spots on the surface of the sample, increased in diameter, indicating the growth of the passive film. When the grooves caused by the surface treatment were not observed, a thick passivation film was formed on the surface of the sample. The surface roughness calculated from the topographic profile was small and changed little, indicating that the passive film was relatively uniform. When the potential reached 0.7 V_(SCE)_, the roughness increased and the passive film became non-uniform, as shown in Figure 2.

In addition, based on the morphology, they found that the passive films contained a mixture of Fe_3_O_4_, Fe_2_O_3_, and FeOOH when passivated at the active–passive transition potential. The inner layer of the passive film was Fe_3_O_4_ and the outer layer was Fe_2_O_3_/FeOOH. When the film-forming potential changed from −0.1 V_(SCE)_ to 0.5 V_(SCE)_, the chemical composition did not change, but the thickness of the inner layer became thicker, leading to an increase in the thickness of the oxide film from about 4.5 nm to 5.8 nm, and the compactness improved, which made the film more protective. On the contrary, when the potential was at 0.7 V_(SCE)_, although the thickness of the film increased, its chemical composition changed greatly and an amorphous structure appeared, which reduced the corrosion resistance of the film.

Padhy et al. [34] investigated the passive film properties of 304L stainless steel in the nitric acid medium in both ex situ and in situ conditions. The ex situ study on the surface morphology of the passive film showed that the surface morphology of 304L stainless steel depended on the stability of the passive film in nitric acid. The morphology features of the passive film in nitric acid medium increased with time and solution concentration. The passive film in 1 M and 4 M nitric acid solution was stable. However, in 8 M and 11.5 M nitric acid, breakdown of the passive film was observed. In situ surface morphology changes of the passive film were monitored by EC-AFM in 0.1 M, 0.5 M, 0.6 M, and 1 M nitric acid. It was found that, at lower concentrations of 0.1 M and 0.5 M nitric acid, the passive film grew in a platelet-like structure. At nitric acid concentrations of 0.6 M and 1 M, platelet-like structures aggregated, homogenized, and began to deplete from the surface, causing selective dissolution. The results of X-ray photoelectron spectroscopy (XPS) analysis showed that the passive film at lower concentration was composed of a hydroxide and oxide layer. However, at high concentrations, the passive film consisted only of an oxide layer. The passivity of 304L stainless steel under the action of low-concentration nitric acid started from the formation of chromium hydroxide, and the surface was in the form of platelet-like structures. As the concentration increased, the hydroxide layer changed to a homogenous oxide layer. As the concentration continued to increase, the protective oxide film was depleted from the structural heterogeneous zone, leading to the opening of the oxidation boundary. The depletion of the oxide layer caused selective dissolution and local corrosion of 304L stainless steel.

Many researchers reported the corrosion process and mechanism of AA 2024 and identified the corrosion products. Kreta et al. [35] explored the changes on the surface during exposure to the sodium chloride environment. In their experiment, the development of the passive film of AA 2024-T3 was analyzed with different analytical tools during exposure to 0.5 M NaCl. In order to track the morphology changes of the protective hydroxide/oxide film, the in situ EC-AFM measurements were carried out on the site without inclusions in a tapping mode. In situ EC-AFM images were taken from samples exposed to different potentials in various time steps. The parameters such as roughness are not only a function of potential, but also a time-dependent function. It can be observed that the surface roughness gradually increased during immersion in sodium chloride solution, and there was a similar increase in surface roughness after the application of external potential, which was mainly due to the formation of an oxyhydroxide layer. In addition, the deposition of the corrosion products led to the formation of a spiky surface, as shown in Figure 3. The profile lines measured over the corroding intermetallic particle are shown in Figure 3e, which allows observing the formation and roughness changes of the corrosion product film. The further increase in potential was most probably responsible for the significant increase in the thickness of the passivation layer. Under a certain potential, the corrosion potential spikes formed but disappeared after the thicker passivating oxyhydroxide layer was formed.

In terms of biomaterials, the inflammatory state of local body fluids in the body may affect the corrosion of implanted metallic biomaterials such as CoCrMo alloys. In the study of Liu and Gilbert, the surface of CoCrMo alloy was imaged by in situ EC-AFM to evaluate the morphology of the oxide film and surface dissolution under different physiologically possible potentials in phosphate buffer saline (PBS) solution and simulated inflammation (SI) solution (PBS solution with 30 mM H_2_O_2_), respectively [36]. EC-AFM images in SI solution showed different topographic changes compared with PBS solution alone, in which the alloy matrix was corroded and the carbide protruded. In contrast, a less compact oxide layer appeared and gradually covered the outermost surface in the case of PBS solution. The authors speculated that the addition of H_2_O_2_ changed or removed the passive oxide films and greatly affected anodic dissolution on CoCrMo alloy. In addition, the grain boundaries and carbide boundaries were preferential parts for the dissolution of oxides, which was mainly due to effects of Cr depletion, a heterogeneous structure, and Cr distribution along grain boundaries and carbide boundaries.

In the work of Bearinger and co-workers, the effects of different potentials and hydration on the properties and structure of passive oxide films on the surface of titanium, 6-aluminum, 4-vanadium (Ti-6Al-4V) were investigated by in situ EC-AFM [37]. The AFM imaging of the different surfaces showed that all samples were covered with a protective titanium oxide dome. Due to hydration, the dome area gradually increased, and coalescence occurred with the increase in applied voltage and time. In addition, Bearinger et al. [38] characterized hydration of titanium/titanium oxide surfaces under freely corroding and potentiostatically held conditions using EC-AFM. In contrast to conventional high-vacuum techniques, EC-AFM enables the measurement of morphological surface structures in the in situ hydrated state. The results showed that the titanium surface covered the oxidized dome and grew laterally during hydration. The applied potential altered the growth rate. Under open-circuit potential, growth proceeded approximately six times faster than under −1 V applied voltage. The oxide growth was partly due to the lateral expansion and overgrowth of the dome at the oxide–solution interface. The method was successfully used to study dynamic changes in the surface morphology.

Metallic glass has excellent mechanical, magnetic, and chemical properties due to its unique structure. It is noted that the corrosion resistance of metallic glass is not only sensitive to its chemical composition but also inseparable from the structure. The influence of the structure on the corrosion of metallic glass is closely related to the dissolution, passivation, and durability of the passive film. Wang et al. [39] selected Ni_50_Nb_50_ metallic glass with polymorphic transition during devitrification to clarify the relationship between the amorphous structure and corrosion, especially the stability of the passive film. By comparing amorphous alloys with its crystalline alloys, the breakdown behavior of passive films and characteristics of the surface film were studied in detail. The surface topography of the samples was characterized by in situ EC-AFM in tapping mode. The results confirmed that the surface of both amorphous and crystalline samples could form metastable pits at nanoscale, but the number of nanometer pits on amorphous samples was much smaller than that on crystalline samples. Moreover, amorphous alloys only had a few deep pits and no shallow pits, as shown in Figure 4a,c, while crystalline alloys had many deep pits and shallow pits, as shown in Figure 4b,d. Although the composition of the passive film on the surface of the two alloys was similar, the amorphous structure could significantly inhibit the formation of pits in corrosive environment. Therefore, amorphous alloys had stronger resistance to pit formation than crystalline alloys.

Phase imaging is an extension of the conventional tapping mode AFM (TM-AFM), through which topographic images and phase images can be obtained simultaneously. The phase images can reflect local nanoscale mechanical properties such as friction, hardness, and viscoelasticity. During the AFM scanning process, different nanometer-scale crystalline phases and their size, shape, and distribution can be identified in the matrix according to the phase change. Zhang et al. [40] studied the formation of the passive film in the Al–Ni–Ce metallic glass by in situ EC-AFM. In situ EC-AFM measurements were performed in open-circuit potential (OCP) conditions and stepped the potential into pitting potential (E_pit_). They identified α-Al nanophase from the amorphous matrix by TM-AFM. It can be clearly seen from the phase image that the small α-Al islands were randomly distributed on the surface, which had a negative phase shift with respect to the matrix. In the partially magnified phase image, the α-Al island is surrounded by bright ringed patches, as shown in Figure 5.

In addition, the nucleation and growth process of the passive film was observed in real time by in-situ EC-AFM, and it was found that the presence of nanocrystalline α-Al precipitates changed the nucleation mechanism of the passive film in the amorphous matrix from instantaneous nucleation to progressive nucleation. Due to the galvanic coupling between the α-Al nanophase (anodic position) and surrounding amorphous matrix (cathodic position), the formation of the corrosion product Al(OH)_3_ was in the early stage of formation of the passivation film, which was mixed into the passive film, changing the local structure and composition of the passive film. The density of hydroxide in the passive film was lower than that of the oxidation film formed on the amorphous matrix, which reduced the compactness of the passive film and its stability. Therefore, compared with fully metallic glass, metallic glass containing nanometer α-Al precipitation had lower corrosion resistance.

Most previous research focused on steady-state corrosion reactions of steel, such as the study of pipeline corrosion in bicarbonate solutions or carbonate bicarbonate electrolytes. However, some researchers also studied the early stage of steel corrosion. Li et al. discussed the influence of bicarbonate concentration on the topographic evolution and corrosion mechanism [41]. In their work, the early stage of X100 pipeline steel corrosion in bicarbonate solution with different concentrations was characterized by in situ EC-AFM in the contact mode. The relationship between surface roughness and the corrosion process of pipeline steel was established. The pipeline steel was fixed to the bottom of the homemade solution container as the working electrode, which prevented evaporation of the test solution during testing. The authors found that the early stage of the steel corrosion in 0.01 M NaHCO_3_ solution could be divided into three stages. When the steel was immersed in the solution, the oxides on the surface of the steel began to dissolve, the OCP dropped, and the surface roughness increased rapidly, representing stage I. When the steel was corroded, the OCP further dropped, and the surface roughness increased gradually, representing stage II. In stage III, as the corrosion of steel reached a steady state, the surface roughness and OCP maintained a stable value, and the formation of corrosion products reached a dynamic equilibrium state. However, in solutions with increased bicarbonate concentration, such as 0.1 M and 0.5 M NaHCO_3_, steel could be passivated. In addition, as the passive capacity of bicarbonate solution increased with the increase in concentration, the morphology of the steel surface in 0.5 M NaHCO_3_ solution was smoother than that in the 0.1 M solution, consistent with the results of power spectral density (PSD) analysis. Compared with the passive film formed at a low concentration (0.1 M), the passive film formed in a solution with a high concentration (0.5 M) could eliminate the large-scale morphological features.

In addition, AFM–SECM is considered to be a particularly attractive research tool in corrosion science [42]. The clear topographical information obtained by EC-AFM and the electrochemical information provided by SECM can be obtained in a single experiment, which makes up for the deficiencies of the two instruments. Izquierdo et al. [43] modified AFM tips to achieve a bifunctional AFM–SECM tip. Conductive microelectrodes were integrated into an AFM probe to obtain an SCEM signal, through which the changes in surface morphology and current density were achieved to observe the progress of corrosion. Local release of Cu^2+^ ions was monitored by electrochemical reduction and deposition of metal ions on the AFM–SECM probe. The formation and breakdown of passive layers were characterized from the perspective of surface roughness and current density, reflecting the initial growth of the passive layer, subsequent breakdown, and formation of the pit.

In the research on microzone corrosion carried out by our research group, Ding studied the corrosion behavior and the formation of **a** corrosion product film on nickel–aluminum bronze (NAB) in 3.5 wt% NaCl solution. The in situ AFM in contact mode at open-circuit potential was used to observe the formation of the corrosion product film on different phases [44]. Figure 6 shows the in situ AFM topography of the specimen surface.

The α phase exhibited different corrosion behavior at different locations. Serious corrosion occurred at the lamellar α phase within the α + κ_III_ eutectoid structure, while the α phase far away from the κ phase was not eroded, mainly due to the formation of many micro galvanic cells in the eutectoid. The κ_II_ and κ_III_ phases exhibited significant corrosion resistance due to the formation of a stable, dense protective film in a short period of time. It is worth noting that the film on the κ_II_ phase was thicker, related to the κ_II_ phase based on Fe_3_Al intermetallic compounds and containing more iron. Compared with nickel oxide, iron oxide is fluffier and more unstable; thus, the film on the κ_II_ phase was thicker. Since the metastable martensite structure struggled to form a protective film, the β’ phase suffered the most severe corrosion. When the immersion time reached 150 min, the corrosion product deposition was almost dispersed on the entire surface, and the corrosion depth of each phase was suppressed, as shown in Figure 7. The formation of the corrosion product film tended to be uniform, which prevented the NAB alloy matrix from contacting the corrosive medium and hindered the transport of ions and charges, as well as improved the corrosion resistance of the NAB alloy.

EC-AFM has great advantages in studying the formation process and microstructure of corrosion product films on a metal surface, especially in characterizing the evolution of early corrosion product films, which is conducive to analyzing the corrosion behavior of metals and promoting the research on the corrosion resistance of metal. The application of EC-AFM in corrosion product films is summarized in Table 2.

#### 3.1.2. Pitting Corrosion

Pitting corrosion was extensively studied in the past few decades, as it has greater destructive and hidden dangers. Generally, the pitting mechanism is related to the properties of impurities on the surface or matrix. Pitting corrosion near inclusions, precipitates, or matrix/impurity interfaces is often caused by the inherent potential difference between the matrix and the inclusions or precipitates. Due to the small anode area, the corrosion rate is very fast, which can lead to sudden accidents. When trying to limit or avoid pitting corrosion, EC-AFM is helpful for improving the understanding of random pitting distribution, including nucleation and growth of pitting.

Pitting corrosion of various stainless steels was the focus of research for many years. As the most used material in the orthopedic and orthodontic bracket, AISI 316 L austenitic stainless steel has good mechanical properties. However, because it is susceptible to localized corrosion in chlorine-containing environments, it is often challenged by corrosive environments in the body. Conradi et al. [45] studied localized corrosion of austenitic stainless steel of the type AISI 316 L and duplex 2205 stainless steel (DSS 2205) using in situ EC-AFM in two different solutions, including simulated physiological solution known as Hank’s solution (PS) and artificial saliva (AS). The topography and surface roughness of DSS 2205 and AISI 316 L changed with the increase in chloride ion concentration. EC-AFM topographic images showed that the surface of DSS 2205 had prominent square-like corrosion products in both solutions. Compared with AS, the erosion of PS was manifested by the decrease in the time scale of DSS 2205 surface change due to the increase in chloride ion concentration, as shown in Figure 8. Even though the DSS 2205 steel was exposed to anodic potentials in the region of transpassive oxidation, the samples had high pitting corrosion resistance in both solutions. On the contrary, AISI 316 L steel had stable corrosion resistance to AS. However, when AISI 316 L steel was exposed to PS solution, the sample was easily corroded and showed obvious pitting corrosion with the increase in Cl^−^ ion concentration, as shown in Figure 9. The surface of AISI 316 L included ellipse-like deposits. This was due to the change in the chemical composition of the matrix material, which formed different growth patterns on the surface of the sample with the growth of chromium, iron, and nickel oxides. Therefore, DSS 2205 had high corrosion resistance compared to AISI 316 L stainless steel. In medical applications, DSS 2205 will be a promising medical material if the nickel hypersensitivity effect can be reduced in patients receiving treatment.

Martin et al. [46] studied the pitting corrosion of austenitic 304L stainless steel in chloride borate buffer solution using in situ EC-AFM in contact mode. In order to determine whether the pits were randomly distributed at the nanometer scale, the study focused on the location where pits were initiated under controlled potential. In the AFM image, it is clear that the chain of pits (black, right image) was related to the chain of relief islands (white, left image), as shown in Figure 10. However, we can see an intermediate state rather than a typical direct transition, which indicates that the pitting process was a gradual process, as shown in Figure 11. In the final step of surface preparation, the local chemical reactivity of the surface may have led to the formation of a passive film and small oxidized hydroxide aggregate defects. The authors believed that the local chemical defects in the sample preparation process could affect the formation of the passive film, which would eventually cause local differences compared to the rest of the film. These local chemicals or structural defects would reduce the local pitting resistance of the film. In addition, based on the reasonable model that the surface potential under tensile stress would be higher than that under compressive stress, when the strain hardening zone (caused by mechanical polishing) appeared on the surface of the sample, pitting corrosion preferentially initiated in this area.

It is generally believed that sulfide inclusions (MnS and mixed oxide/sulfide) in a stainless-steel matrix are most likely to cause pitting corrosion. The products produced by the dissolution of sulfide inclusions form a local corrosive environment, which in turn causes more severe pitting corrosion. Wijesinghe et al. [47] discussed the adverse effect of sulfide inclusion on pitting resistance of stainless steel. The results showed that the sulfide inclusions were clustered on the surface of the stainless steel. As the volume composition of the stainless-steel sulfide increased, the number of inclusions per cluster increased. The pitting corrosion was imaged in real time using in situ EC-AFM, and it was found that pits were formed near the sulfide inclusions, consistent with the three main mechanisms proposed previously, i.e., aggressive local chemical reactions based on inclusion dissolution, stressed oxide, and chromium depletion.

In the work of Zhang and co-workers, the pitting corrosion of the solution- and sensitization-treated austenitic stainless steel SUS304 was studied using in situ AFM in 3.5 wt% sodium chloride solution [48]. This study adopted two observation methods. One was in situ continuous observation, in which the corrosion current was continuously applied to the surface of the sample. Another method was in situ interrupted observation, i.e., the observation was carried out at intervals to minimize the impact of probe scanning on corrosion reactions. In situ observation of the corrosion pit of the solution-treated sample showed that the pit became larger as the corrosion time increased from t = 2.46 to 2.70 ks, but no corrosion products were found, as shown in Figure 12. A series of in situ intermittent observations of the sample after solution treatment revealed two large pits (D and E), but the size of the pits did not change with time. In addition, no corrosion products were found on the two large pits, as shown in Figure 13.

The authors believed that the corrosion product may have been moved to other locations by the probe or dissolved with the rapid dissolution of the matrix. After the corrosion product was removed, the concentration of local chloride ions and hydrogen ions decreased, and the growth rate of the pits decreased. This means that the corrosion products played an important role in the growth of corrosion pits. When the corrosion product covers the pit, it can lead to an acidic environment and accelerate pitting. On the sensitization-treated sample, the irregular pits were distributed near the grain boundaries. Chromium carbide deposits and pitting occurred in the chromium-depleted area around the carbide, as shown in Figure 14. Cross-sectional profiles of pits along a–b lines in Figure 14a and c–f lines in Figure 14c are shown in Figure 14d. Since no carbide particles were found in the pit according to the cross-sectional profiles, they might have dissolved with pit growth. The dissolution of carbides in pits caused the pit to grow further.

When we study the corrosion of self-passivation metals covered with a semiconductor protective film, there is no doubt that EC-AFM, which does not need the conductivity of the matrix, is more suitable. Qu et al. [49] investigated the corrosion behavior of pure aluminum using in situ AFM in 0.01 mol/L FeC1_3_ solution. The pitting corrosion process of pure aluminum induced by the potentiodynamic sweep and the repassivation process of active pits were studied. In addition, the effect of mechanical damage on the metal surface caused by AFM tip scratching on the pitting behavior of the sample was emphasized. The topographical images of the sample surface at different immersion times were traced under open-circuit conditions using AFM in contact mode, and the AFM tip scratching process was carried out with a loading force of 800 nN. The results showed that different pitting regions exhibited different pitting activities under the same polarization conditions due to the diversity of physical and electrochemical characterization. The corrosion products contained abundant impure elements such as Fe and Cu. In situ AFM observation of pitting corrosion originated from artificial defects on the aluminum surface showed that physical defects had higher pitting activity and may have been attacked preferentially to pitting corrosion.

In the study of Davoodi and co-workers, the application of the EC-AFM and SECM integrated system for in situ studies of the influence of intermetallic particles on local corrosion of aluminum alloys was introduced [50]. The key to this method is fabricating a dual-mode cantilever/tip that can be used not only as a cantilever for the EC-AFM but also as a microelectrode or nano-electrode tip for the SECM. It can obtain the in situ AFM topographic images and SECM electrochemical current maps simultaneously with micrometer lateral resolution and provide detailed information of localized corrosion related to different kinds of intermetallic particles and the deposition of corrosion products. In their work, when the AA1050 sample was anodically polarized at 300 mV (close to the breakdown potential), the morphology observed using EC-AFM showed many small pits. However, this may have been because many of the active points were close to each other; thus, the local current could not be resolved in the SECM electrochemical current map. During the post-scan, some high-current sites appeared in the lower part of the SECM image, which may have been related to the presence of pits or trenches on the surface, as shown in Figure 15. Preliminary results indicated that the localized dissolution of aluminum alloy may have occurred below the breakdown potential, but only involved a small number of particles.

Metallic zinc is used not only for galvanizing steel but also for various applications such as batteries, die casting, and brass metallurgy. It is necessary to study the local corrosion of pure zinc and zinc alloys. Amin et al. [51] studied the passivation breakdown and pitting corrosion of zinc in 0.5 M sodium hydroxide solution containing different concentrations of ClO_3_^−^ or ClO_4_^−^ anions. The passivation breakdown and pitting sensitivity of the Zn/OH^−^/ClO_3_^−^ and Zn/OH^−^/ClO_4_^−^ interfaces were studied by potentiodynamic anodic polarization measurements, and the topography of pitted Zn surfaces was observed by AFM. The experimental results showed that metastable pitting corrosion was mainly caused by Cl^−^ ions (generated by reducing ClO_3_^−^ and ClO_4_^−^). According to the point defect model (PDM), this may have been due to the small size of Cl^−^ ion, which could occupy anion vacancies and cause pitting corrosion. Therefore, Cl^−^ was the invasive ion that caused metastable pitting, rather than ClO_3_^−^ and ClO_4_^−^. However, even in the absence of Cl^−^, ClO_3_^−^ and ClO_4_^−^ could induce stable pitting, in which ClO_3_^−^ was more aggressive than ClO_4_^−^, which could be proven by the roughness calculated according to the AFM topography map. Compared with ClO_4_^−^, the Ra value was always higher in the presence of ClO_3_^−^, as shown in Figure 16.

Yong Hwan Kim et al. [52] studied the initial corrosion mechanism of a hot-dip-galvanized surface. The formation of corrosion products, the initiation and growth of pits, and the breakdown of the film were observed using in situ EC-AFM. At the initial stage of passivation, the corrosion product film of ZnO/Zn(OH)_2_ formed in the dull sector was more unstable than that formed in the bright zone. The authors believed that the relative instability of the passive film in the initial stage was affected by the high-density lath-like structure. The uneven surface structure at the micro-scale, i.e., the lath-like structure, provided a favorable place for the ZnO/Zn(OH)_2_ formation and pitting.

EC-AFM can provide detailed information of the localized dissolution associated with different kinds of intermetallic particles, as well as the deposition of corrosion products surrounding large particles or covering small pits, including the location of pitting initiation at controlled potentials, and whether pits are randomly distributed at the nanoscale. The application of EC-AFM in pitting corrosion is summarized in Table 3.

#### 3.1.3. Selective Corrosion

Selective corrosion refers to the preferential dissolution of active components in the multicomponent alloy. The relative Volta potential difference is used as the index of selective corrosion caused by microelectronic coupling. Compared with the substrate, the intermetallic phase with positive potential is the cathodic phase. On the contrary, some intermetallics with more negative potential than the matrix show stronger activity during the corrosion process and act as micro-anodes.

Jia et al. [53] investigated the effects of intermetallics on the local corrosion behavior of AZ91 alloy added (La,Ce) mischmetal (MM), including corrosion initiation and propagation. The corrosion morphologies of different intermetallics were imaged using in situ EC-AFM in 0.1 mol/L NaCl solution, and Scanning Kelvin probe force microscopy (SKPFM) was performed to measure the local difference of the relative Volta potential between the intermetallic phases and the α-Mg matrix. In the experiment, the approximate compositions of several intermetallic phases were identified as Al_4_(La,Ce), Al_8_Mn_4_Ce, and decreased b-Mg_17_Al_12_ phases. SKPFM analysis showed that all intermetallics were noble compared with the α-Mg matrix. The observation of in situ EC-AFM on the initial stages of corrosion indicated that the a-Mg matrix surrounded by b-Mg_17_Al_12_ or Al_8_Mn_4_Ce was prone to pitting corrosion. In detail, the b-Mg_17_Al_12_ phase existed in both AZ91 and AZ91 alloys with (La,Ce) MM. Both elongated and dispersed granular b-Mg_17_Al_12_ served as effective micro-cathodes, as shown in Figure 17.

Al_4_(La,Ce) intermetallics, whether featuring an acicular shape or rod shape, were believed to produce significant galvanic coupling. However, the α-Mg matrix near the Al_4_(La,Ce) phase did not have obvious corrosion. Moreover, although the A_l8_Mn_4_Ce phase showed a positive potential difference with respect to the matrix, the corrosion resistance of the alloy was not affected by its galvanic coupling, probably due to its lower amount, as shown in Figure 18.

It can be seen that the intermetallics with the highest Volta potential differences relative to the matrix did not play the role of effective cathodes, just like the Al_4_(La,Ce) phase, which means that the localization of the cathode reaction was not only dependent on the Volta potential differences between the intermetallics and the matrix. The authors believed that the shape of the coupled anode and cathode had an important effect on the current density, i.e., the geometry of the intermetallic compound had an important influence on the micro-galvanic corrosion.

Zhang et al. [54] studied the anodization process of aluminum 6060 alloy under operating conditions and illustrated the effects of intermetallic particles (IMPs) and anodic Al oxide (AAO) film properties. The in situ EC-AFM measurement was performed continuously to monitor the surface topography changes under anodizing potentials in contact mode, revealing the details of localized dissolution and AAO film formation in Al 6060 samples in 0.2 M Na_2_SO_4_ solution. The extruded Al 6060 alloy mainly contained two types of IMPs: AlFeSi primary particles and Mg_2_Si particles. The Volta potential differences obtained by SKPFM showed that, relative to the aluminum matrix, AlFeSi was cathodic, but Mg_2_Si was anodic. The in situ EC-AFM measurements showed that AlFeSi particles remained stable, but the local anodic dissolution of Mg_2_Si particles occurred during anodization, which was consistent with the SKPFM results. As shown in Figure 19, the large particle in region I was likely an α-AlFeSi particle. The particles in regions II, III, and IV were likely anodic Mg_2_Si particles. The protruded large particle in region I remained stable during anodization, whereas the small particles in region II in Figure 19a were dissolved upon the anodization. Moreover, upon the anodization at 1 V, some active dissolution started at certain sites (e.g., III), forming small holes, as shown in Figure 19b. During the anodization at 2, 4, and 8 V (Figure 19c–e), the small hole in area III became deeper, as displayed by the profile lines in Figure 19f, and pronounced localized dissolution occurred in area IV. Two small particles remained stable in the dissolved area (area IV in Figure 19d,e). Furthermore, in the area marked as V, localized dissolution at one site resulted in a small hole, and a deposited particle of a few μm in size formed (Figure 19c), but it disappeared at 4 V, exposing a much deeper pit (Figure 19d). In addition, the growth of AAO films occurred with partial anodic dissolution. The thickness of the anodic barrier film increased linearly with the anode potential, but the growth rate decreased due to local anodic dissolution associated with IMPs in the alloy.

In some cases, the presence of nitrides can adversely affect the corrosion resistance of the material. Bettini et al. [55] studied the effect of nano-sized quenched-in chromium nitride particles on the corrosion behavior of heat-treated 2205 duplex stainless steel (DSS) in an NaCl solution at room temperature and 50 °C (slightly higher than the critical pitting corrosion temperature). The relative nobility difference between the precipitated nitrides, austenite, and ferrite in the tested materials was evaluated at room temperature by atomic force microscopy-based Kelvin force microscopy (AFM/KFM). The volt potential mapping at room temperature indicated that the ferritic had a lower relative nobility compared with austenite, and quenched-in Cr_2_N particles had a higher relative nobility to the surrounding ferritic matrix. The observation results of EC-AFM in 1 M NaCl solution at room temperature showed that the samples after heat treatment showed a wide range of passivation and a very stable surface up to 1.2 V_Ag/AgCl_, where the selective dissolution of ferrite phase occurred but the quenched-in Cr_2_N particles remained stable, as shown in Figure 20. Figure 20d shows the depth line profile from the image obtained after the polarization at 1.2 V_Ag/AgCl_, presenting a depth of ca. 200 nm between the dissolved ferrite phase and the remaining austenite phase. In addition, the very small nitride particles formed during the fast cooling process did not have enough time for the diffusion of elements and were unlikely to form a considerable composition gradient in the surrounding boundary region, which was one of the reasons for its stability. At temperatures above the critical pitting temperature (CPT = 50 °C), rapidly selective dissolution of the austenite phase occurred upon slight anodic polarization, which may have been related to the low content of chromium and molybdenum in the austenite phase. The authors believed that the finely dispersed quenched-in nitrides in the DSS did not cause local corrosion in 1 M NaCl solution. However, the exposure temperature had a great influence on the corrosion resistance of DSS, which changed the selective dissolution behavior of DSS.

Yasakau et al. [56] investigated the mechanism of initial steps of localized corrosion at the cutting edges of adhesively bonded Zn (Z) and Zn–Al–Mg (ZM) galvanized steel substrates. The topography of the initially localized corrosion of Z and ZM samples was measured using in situ AFM equipment under anodic polarization in the corrosive solution. The first corrosion pits at the Z cutting edge were mainly formed on the zinc layer, and there was no preferential erosion on the adhesive/zinc or zinc/steel, as shown in Figure 21.

However, the types of attacks were different at the ZM cut edge, namely, pitting corrosion of the solid solution and selective dissolution at the eutectic zone. For the local corrosion at the adhesive/zinc interface, the first adhesive disbonding area was located near the cutting edge. The second adhesive disbonding zone was located at the buried deep zinc/adhesive interface, where local corrosion occurred in the solid solution phase and eutectic phase of the sample ZM. Under anodic polarization, the corrosion site was located in the eutectic phase at the interface between the eutectic zone and the solid solution zone, similar to the corrosion at the ZM cutting edge, as shown in Figure 22. In addition, scanning vibrating electrode technology (SVET) and electrochemical impedance spectroscopy (EIS) electrochemical tests showed a decrease in corrosion kinetics at the Z and ZM cutting edges, which was due to the blocking effect of the dense film of corrosion products formed on the zinc and steel surfaces.

Depentori et al. [57] studied the corrosion behavior of neodymium-modified titanium alloy Ti_6_Al_4_V_2_Nd in 1.5 wt% NaCl and compared it with a Ti_6_Al_4_V_2_ matrix using SKPFM and EC-AFM. Neodymium containing intermetallic compounds was precipitated at β-phase boundaries and inside the grains. The Volta potential maps obtained using SKPFM showed that the Volta potential of intermetallic compounds was lower than that of the Ti matrix. Therefore, intermetallic compounds had strong anodic behavior relative to titanium matrix and were the preferred site for local attack in Ti_6_Al_4_V_2_Nd alloy. When the sample was immersed in 1.5 wt% NaCl solution, EC-AFM observation showed that the volume of the local area on Ti_6_Al_4_V_2_Nd increased significantly, which was due to the formation of hydroxide and oxide. In addition, large amounts of debris were observed. The authors suggested this as a sign that the surface was loosely bound to Nd(OH)_3_. Cyclic voltammetry showed a clear oxidation reaction without a reduction reaction, and the oxidation peak moved to the right as the exposure time increased, which was a clear sign that the electrochemical reaction was irreversible and that the diffusion barrier formed and increased over time. Corresponding to the results of EC-AFM, the diffusion barrier was the Nd(OH)_3_ layer formed by the corrosion of a large number of precipitates on the surface, which prevented further oxidation.

Davoodi et al. [58] investigated the difference in corrosion behavior between EN AW-3003 (Rolled 3xxx series Al alloys) and a newly developed Al–Mn–Si–Zr fin alloy. The Volta potential of the two alloys determined using SKPFM showed that the intermetallic particles behaved as cathodes relative to the alloy matrix. Compared to EN AW-3003, the Al–Mn–Si–Zr alloy had fewer particles with larger Volta potential difference with respect to the matrix. In situ AFM measurements showed that ring-like corrosion products were deposited on the EN AW-3003 alloy, while only a few corrosion sites and tunnel-like pits were found on Al–Mn–Si–Zr, as shown in Figure 23.

Topography and electrochemical currents obtained synchronously by the integrated SECM/AFM systems provided information on the pitting precursor and pitting process. Compared with Al–Mn–Si–Zr alloys, EN AW-3003 alloys had more active sites and extensive localized dissolution, resulting in higher material losses. Interestingly, some of the larger micron-sized intermetallic particles initiated localized dissolution at the boundary region of the particle-matrix, while the fine dispersions were not active. The intermetallic particles in the Al–Mn–Si–Zr alloy were few. Although they could induce selective dissolution and form small tunnel-like holes, because of their small weight loss, they were suitable for fin material in heat exchange applications.

The combination of EC-AFM and scanning Kelvin probe force microscopy (SKPFM) can simultaneously obtain topographical changes and Volta potential maps, which helps to better understand selective corrosion behavior and its mechanism. The application of EC-AFM in selective corrosion is summarized in Table 4.

#### 3.1.4. Intercrystalline Corrosion

Intercrystalline corrosion is a local corrosion phenomenon that occurs along the grain boundary of a metal material in a corrosive medium and causes the loss of bonding force between grains. When intercrystalline corrosion occurs in the material, there may be no macroscopic change, but the material’s strength is almost completely lost, resulting in the sudden destruction of equipment. The main reason for intercrystalline corrosion is the difference in structure and chemical composition between the grains and grain boundaries of the material.

Based on EC-AFM, the contact mode high-speed AFM (HS-AFM) compensates for the shortcomings of AFM with short collection times. The long collection time is a limiting factor for AFM. Contact mode HS-AFM images multiple frames per second, making it orders of magnitude faster than traditional AFM. This enables real-time imaging processes with nano-scale lateral resolution and sub-nanometer-scale height resolution. The increase in speed can not only directly image dynamic nanoscale events, but also macroscopic regions of the sample surface without reducing resolution. It is a valuable imaging tool for in situ observation of nanoscale corrosion initiation events such as metastable pitting, grain boundary dissolution, and short crack formation during stress corrosion cracking.

In the work of Moore and co-workers, local corrosion phenomena such as pitting and intergranular attack (IGA) on thermally sensitized AISI 304 stainless steel were studied using HS-AFM in 1% NaCl [59]. Real-time in situ HS-AFM observations showed that an intergranular pit was formed within 0.5 s during a galvanostatic scan. Intergranular pits were distributed along the grain boundary (GB) in a chain shape, which was caused by the preferential corrosion of GB. Chromium carbide was precipitated along GBs, which resulted in the depletion of the local chromium elements in the area around GBs, greatly reducing the local corrosion resistance of GBs. By using HS-AFM and electrochemical data for the computational model, it was found that the radial diffusion state of the system was reached within 0.01 s, leading to rapid dissolution of materials.

Padhy et al. [60] used EC-AFM to study the surface morphology of austenitic stainless steel in nitric acid medium. In situ EC-AFM results showed that the surface presented a platelet-like structure in low-concentration nitric acid solution (0.1 M, 0.5 M), providing effective protection for the surface. When the concentration of HNO_3_ was from 0.1 M to 0.6 M, the roughness decreased, which was related to the thinning of the passive film and marked the beginning of corrosion. From the morphology of 0.6 M HNO_3_, breakdown of the passive film and surface dissolution were observed. As the concentration continued to increase to 1 M, the roughness increased due to the intensification of the surface dissolution and selective dissolution of grain boundaries, as shown in Figure 24.

The early stages of localized corrosion are important as mentioned earlier, because they account for most of the lifetime of intergranular corrosion, stress corrosion cracking (SCC), or pitting, and they are more accessible to take remedial action. Williford et al. [61] used EC-AFM to obtain images of the early stages of intergranular corrosion (IGC) in 304L stainless steel. The observation of EC-AFM showed that IGC was present between the carbides, but not completely around the carbide in the early stage. Later, the grain boundaries became wider, the carbides were shortened, and the IGC completely surrounded the carbides. The matrix between carbides began to corrode, which was best explained by the fact that the carbides were cathodic with respect to the matrix; thus, when the matrix dissolved, they were protected by cathodic protection. In addition, the carbide was connected to adjacent grains through the ligaments of the matrix material, which may have meant that the SCC crack front was bridged by the carbide. The SCC crack front could move forward and surround the carbide, creating an area where the carbide bridged the crack.

An AFM with a conductive probe can simultaneously obtain the surface topography and surface potential of the scanned surface. Fu et al. [62] studied the local corrosion of high-chromium cast iron in regions at different distances from interphase boundaries using EC-AFM in the contact mode. According to the measured local potential, a decrease in interface potential between carbide and matrix was observed. Moreover, the corrosion rate of the metal matrix near the primary carbide was significantly higher than that far away from the primary carbide. The morphology map obtained using EC-AFM showed that the area near the primary carbide/matrix interface corroded or dissolved more rapidly than the area away from the interface. Furthermore, the corrosion rate was found to be particularly rapid in areas with sharp edges. The authors suggested that, on the flat side, most of the electrons flowed into the half-space on one side of the matrix. However, at the sharp edge, electrons flowed into a larger space in the matrix, corresponding to a larger total electron flow, which greatly increased the corrosion rate of the sharp edge.

In high-entropy alloys (HEAs), the homogeneous elemental distribution is expected to improve corrosion resistance. In order to develop highly corrosion-resistant HEAs, it is necessary to study the relationship between chemical/microstructure segregation and localized corrosion. In the study of Shi and co-workers, localized corrosion of Al_x_CoCrFeNi HEAs was studied using in situ EC-AFM in 3.5 wt% NaCl solution. Surface topography changes at micron/submicron scales were monitored under different anodic potentials [63]. In the experiment, with the increase in aluminum content in HEAs, the microstructure changed from a single FCC (face-centered cubic) solid solution to the FCC phase and (ordered/disordered) BCC (face-centered cubic) phase. The EC-AFM image showed that the uniform single-phase Al_0.3_CoCrFeNi alloy had the best corrosion resistance, and the breakdown of its passive film was in the form of randomly distributed pits, as shown in Figure 25.

With the increase in aluminum content, the BCC phase appeared in the Al_0.5_CoCrFeNi alloy, resulting in a heterogeneous microstructure. Pits were formed preferentially along the FCC/BCC phase boundary, leading to the initial breakdown of the passive film, which was manifested as the decline in critical pitting potential value in the potentiodynamic polarization curve, as shown in Figure 26.

Pitting corrosion was not observed in the Al_0.7_CoCrFeNi alloy with the increase in aluminum content and volume fraction of the BCC phase. In contrast, local corrosion along the dendritic/interdendritic boundaries and selective dissolution of the (Al, Ni)-rich ordered BCC phase occurred, as shown in Figure 27. Therefore, the author believed that, as the microstructure changed from single solid solution to multiphase, the breakdown of the passivation film changed from pitting to phase boundary dissolution, which led to a decrease in corrosion resistance.

Bettini et al. [64] studied the effects of carbides on the corrosion/dissolution behavior of biomedical CoCrMo alloys in PBS solution using EC-AFM at different applied potentials. SKPFM results showed that, compared with the matrix, the Volta potential of carbides was higher. In addition, the Volta potential decreased in the boundary region, which may have been related to local depletion of the main alloy elements. This indicated that the carbide boundary had a greater corrosion tendency and was the preferred site for corrosion/dissolution. In situ EC-AFM measurements of the CoCrMo alloys exposed to PBS showed that, at high anodic potential, a dissolution process at carbide boundaries was observed, and an increase in boundary depth was seen in line profiles across these boundaries, as shown in Figure 28. This was consistent with the SKPFM Volta potential mapping, which showed that some of the boundary areas were weak sites for corrosion/dissolution due to lower relative nobility compared to the matrix.

Davoodi et al. [65] studied the localized corrosion and preferential dissolution of Al alloys in chlorine solution using an integrated EC-AFM/ SECM system. The integrated EC-AFM/SECM could simultaneously detect topographic changes and electrochemical active sites in the same region and reveals local corrosion processes related to IMPs. The results showed that preferential dissolution occurred in the interfacial region between the alloy matrix and IMPs. The formation of grooves around the larger IMPs indicated that different types of IMPs had different dissolution behaviors. In addition, they found that only a small number of IMPs were involved in the localized dissolution at any given time.

EC-AFM can clearly reveal the formation of trenches and the local dissolution of the grain/phase interface, and it can explore the causes and mechanisms of intercrystalline corrosion in combination with other test tools. Table 5 summarizes the application of EC-AFM in intercrystalline corrosion.

### 3.2. Metal Protection

#### 3.2.1. Coating Protection

One of the primary protection techniques for metallic materials is the use of a cover layer on the metal surface to avoid direct contact between the metal and the corrosive medium as much as possible. The coating provides an effective barrier to the substrate, slowing the corrosion of the metal. In general, the metal surface coating can be divided into the metal coating and non-metallic coating. The coating not only slows electron transfer between the anode and cathode; it can also act as a barrier to prevent oxygen from penetrating the cathode reaction. Microcrystalline coating, nanocrystalline coating, gradient coating, composite coating, etc. can effectively improve the overall performance of the material, including corrosion resistance.

Li and co-workers studied the electrochemical mechanism and corrosion protection properties of solvent-borne alkyd composite coating containing 1.0 wt% CeO_2_ nanoparticles (CeNPs) and 1.0 wt% polyaniline (PANI) for carbon steel in NaCl solution [66]. In their work, the morphology changes of the coatings and redox reactions of PANI at the nanoscale were accurately monitored by linking the volume changes observed using in situ EC-AFM imaging with redox peaks measured using in situ cyclic voltammetry (CV). The results of EC-AFM showed that PANI nanoparticles in the alkyd matrix exhibited contracted morphology in the reduced state, leucoemeraldine base (LB), and expanded morphology in the oxidized state, emeraldine salt (ES). The surface did not change significantly, which indicated that the composite coating was stable in corrosive solutions even under very harsh potential conditions, as shown in Figure 29. OCP and EIS results indicated that the redox reaction of PANI ES/LB forms caused metal passivation, which was an active corrosion protection mechanism.

The evolution of OCP with exposure time under 3.0 wt% NaCl for composite coating and reference coating showed that the potential value of composite coating was higher than that of reference coating during the whole measurement process, indicating that the corrosion resistance of the composite coating was improved. The increase in OCP value was attributed to the fact that CeNPs significantly improved the barrier effect and slowed down the migration of corrosion ions. Therefore, the synergistic effect of PANI and CeNPs greatly improved the barrier performance and the corrosion resistance of alkyd composite coating.

In a series of studies by Li and his colleagues, they also studied the electrochemical activity of 1.0 wt% *p*-toluene sulfonic acid (PTSA)-doped PANI in solvent-borne alkyd composite coating, as well as its self-healing corrosion protection mechanism on carbon steel in 3.0 wt% NaCl solution [67]. Through the CV (cyclic voltammetry) curves and EC-AFM imaging, it was proven that doped PANI at low percentage had extremely high electrochemical activity, showing contraction at the reduction potential and expansion at the oxidation potential, which provided evidence for a reversible redox reaction between the ES and LB forms. The voltage potential diagram obtained using KFM under air conditions indicated that PTSA-doped PANI had sufficiently high electrochemical activity and stable reoxidation ability to keep the passivation region on the metal surface. The PTSA-doped PANI in alkyd composites caused structural mutations due to energy input, which increased the electrochemical activity and provided doped PANI with a good electrochemical connection to the metal surface. It played an important role in improving the self-healing corrosion protection of composite coatings. In addition, the EIS showed increased resistance of the composite coating, which may have been related to the interaction of PANI particles in the alkyd matrix forming a dense and more resistive network.

Moreover, they studied the corrosion protection properties of a waterborne acrylic composite coating with 1.0 wt% acetic acid-stabilized CeNPs on carbon steel in 3.0 wt% NaCl solution [68]. The results of in situ AFM showed that the CeNPs embedded in the composite coating could greatly reduce the nano-pinholes in the waterborne acrylic coating, as well as significantly improve the stability of the coating, which played an important role in improving the barrier property of the coating. In situ EC-AFM indicated that some CeNPs and aggregates were released from the coating surface during exposure, and then some particles and cerium compounds were precipitated, as shown in Figure 30. The presence of CeNPs or aggregates acted as nucleation sites to promote precipitation on the coating surface and inside the coating pinholes, thereby preventing the entry of corrosive ions and the corrosion of the metal matrix.

Liu et al. [69] studied the pitting behavior of austenitic stainless steel with nanocrystalline (NC) and polycrystalline (PC) microstructures in 3.5 wt% NaCl solution. In situ AFM was used to study the process of passive film formation on the PC alloy and NC alloy under anodic potential. AFM observations showed that the passive film formed rapidly on the PC alloy, and pitting occurred after continuous film formation, which was a slow process of metastable pitting formation and reparation. However, the formation of a passive film on the NC alloy indicated that, due to the accumulation of many small particles on the surface, the oxide particles grew in the original position and eventually became a passive film. The voids and the boundaries of oxide particles may have been inoculation points for metastable pits. Although metastable pits were easy to initiate on the NC coating, the small grain size promoted the diffusion of elements, such that pits could be quickly repaired or healed. Therefore, the pitting mechanism of the NC coating was mainly characterized by rapid metastable pitting initiation and death, and its pitting resistance was higher than that of the PC alloy. In addition, in their other study, the characteristics of both pit initiation and pit growth processes on an austenitic stainless-steel NC coating were monitored using in situ AFM [70]. Pit initiation included the formation of metastable pits and repassivation process. Pitting growth included stable pit growth and material dissolution. The fine grain size promoted the formation and growth of nano-scale oxide particles, which significantly improved the repassivation ability and reduced the probability of stable pit formation. Compared with PC austenitic stainless steel, nanocrystals promoted the formation of metastable pits but reduced the rate of stable nucleation and growth of pits.

In the study of Pan and co-workers, the pitting corrosion of coarse crystal (CC) 304 stainless steel and its NC thin film was studied in 3.5 wt% NaCl solution, especially the influence of nanocrystalline on the pitting process [71]. The whole pit growth process was recorded using in situ AFM and the growth mechanism of stable pits on NC film was understood. The results showed that the initiation site of the pit on the NC film was at the boundary of the oxide particles. As there were lots of boundaries on the surface, metastable pit events on NC films were more likely to occur than those on CC 304 stainless steel, which indicated that nanocrystallization promoted metastable pit processes, as shown in Figure 31.

In addition, the transition from metastable pitting to stable pitting was inhibited due to the excellent repassivation ability of NC films. In the process of NC film deposition, the internal residual stress may have inhibited the formation of lace cover in the process of stable pit growth, and then changed the growth mechanism of the NC film surface stable pit. Therefore, the probability of developing from metastable pitting to stable pitting on NC films was much lower than that on CC 304L stainless steel. Nanocrystallization changed the geometry and growth mechanism of stable pits, slowing down the nucleation and growth process, which improved the pitting corrosion resistance of CC 304L stainless steel. Moreover, they also studied the corrosion behavior of a magnetron-sputtered NC 304L stainless steel coating in 0.05 M H_2_SO_4_ + 0.2 M NaCl solution, which was compared with conventional rolled CC 304L stainless steel [72]. The nanocrystalline structure reduced the adsorption capacity of Cl^−^ on the surface and inhibited the incorporation of Cl^−^ in the passive film. In situ AFM observation showed that the growth rate of the passive film on the NC film was greatly higher than that on the CC 304L stainless steel, as shown in Figure 32. In other words, the nanocrystalline structure improved the growth rate of the passive film and facilitated the healing of the passive film rupture. The composition of the passive film on the NC film determined by XPS had a higher ratio of chromium oxide to iron oxide. The higher content of chromium oxide improved the corrosion resistance of nanocrystalline samples. Since the structure of the passive film was more compact, the ratio of chromium oxide to iron oxide was higher, and the incorporation of Cl^−^ was less, the corrosion resistance of the NC film was greatly improved.

In addition, the passive film growth mechanisms of the NC 304L stainless-steel thin film, deep-rolled bulk nanocrystalline (BN) 304 stainless steel, and CC 304 stainless steel in 0.05 M H_2_SO_4_ + 0.2 M NaCl solution were studied using electrochemical measurements and in situ AFM [73]. The growth rate of the passive film on the three materials was in the following order: NC thin film > BN304 stainless steel > CC 304L stainless steel. Nanocrystallization changed the nucleation mechanism of passive films from gradual to instantaneous. The passive film on the CC 304L stainless steel and BN 304L stainless steel had a single-layer structure, while the passive film on the NC film had a multi-layer structure.

EC-AFM can not only detect the state of the coating surface through high-resolution imaging, but also produce coating defects by means of probe scraping to obtain direct information on the coating corrosion resistance. The application of EC-AFM in coating protection is summarized in Table 6.

#### 3.2.2. Corrosion Inhibitor Protection

A corrosion inhibitor is a chemical or a mixture of several chemicals that prevents or slows corrosion when present in a corrosive environment (medium) in the proper concentration and form. The addition of corrosion inhibitors can significantly reduce the corrosion rate of metal materials. At the same time, the original physical and mechanical properties of the metal material can be maintained. The advantages of using corrosion inhibitors to protect metal lie in their low dosage, quick effect, low cost, and convenient use. At present, corrosion inhibitors are widely used in machinery, petrochemical, metallurgy, energy and other industries. In some studies, the authors used in situ EC-AFM under real-time operating conditions to detect changes in the corrosion morphology of the sample after the addition of the corrosion inhibitor, and then investigated the properties of the corrosion inhibitor and speculated on the corrosion inhibition mechanism.

The study of non-toxic corrosion inhibitors is important for replacing classical molecules with sulfur, nitrogen, or aromatic functions. Rocca et al. [74] reported the inhibition conditions and mechanisms of linear sodium heptanoate on copper corrosion. In situ EC-AFM under applied potential allowed observing the morphology of the passive film without damaging the layer. The result showed that, when pH = 5.7 and 11, for 0.08 M NaC_7_, large non-covering copper heptonate crystals or non-covering copper oxide were formed on the surface, and the corrosion inhibitor efficiency was low in acidic medium. However, when the pH value was 8, for 0.08 M NaC_7_, a thin layer of heptanoic was formed, which acted as a barrier and effectively protected the metal matrix. The inhibition of sodium heptanoate was related to the formation of a protective layer consisting mainly of copper heptanoate on copper, and the optimum corrosion inhibition conditions were 0.08 M NaC_7_, pH 8 for copper corrosion.

Bertrand et al. [75] used in situ EC-AFM to study the corrosion behavior of a copper surface immersed in various electrolytes under dynamic potential conditions at room temperature. In sodium sulfate and sulfuric acid solutions, dissolution precipitation occurred on the surface, which changed the topographic characteristics of the metallic surface. On the contrary, when copper was immersed in borate or heptanoate solutions, passivation could be clearly observed. Regardless of the oxidation mode (constant potential or open-circuit condition), a thin passive layer was grown on the surface and was stable over time. In the borate medium, the sediments were composed of Cu_2_O and CuO oxides, while, in the heptanate electrolyte, a metal soap composed of copper(II) heptanate was detected. In addition, the inhibition mechanism of sodium heptanate was identified, whereby a thin passive layer of copper metal soap was formed on the surface via dissolution precipitation.

The use of polymer corrosion inhibitors attracted attention. On the one hand, they have low cost and good stability. On the other hand, they have multiple adsorption sites that form complexes with metal ions covering the surface and protecting the metal from corrosion. Umoren et al. [76] studied the mechanism of polyacrylic acid (PAA) inhibiting the corrosion of pure cast aluminum in acidic medium and the synergistic effect with iodide ion addition. The in situ AFM morphology of the surface showed that PAA was adsorbed onto the surface of the aluminum and its arrangement was more orderly in the presence of iodide ions, as shown in Figure 33, resulting in higher inhibition efficiency. The authors believed that the alumina film was replaced by the adsorption of KI on aluminum, and then the PAA was adsorbed onto the KI such that the PAA molecules were arranged on the aluminum surface in an orderly manner, which enhanced the inhibition process.

Zhang et al. [77] investigated the electrochemical corrosion behavior of AISI321 stainless steel in 36% ethylene glycol–water solution. It can be seen from the results of EC-AFM that the passive film became more complete and the number of defects decreased with the increase in polarization time. Moreover, as the passive potential increased, the particle diameter increased and surface defects decreased. When the potential ranged from −0.15 to 0.45 V, an N-type oxide film adhered to the surface of the sample. However, when the potential was between 0.45 and 0.75 V, a P-type oxide film was formed on the surface. The passive film formed at high potential had fewer defects and excellent protective performance. Therefore, AISI321 stainless steel could be passivated in ethylene glycol solution and the passive potential ranged from −0.15 V to 0.75 V. With the increase in passive potential, the protective performance of the passive film was significantly improved.

In the study of Nikhil and co-workers, ethyl-2-cyano-3-(4-(dimethylamino) phenyl) acrylate (ECDPA) and ZnO nanosheet composites were synthesized and used as corrosion inhibitors for copper in 1 M HCl. ECDPA acted as a barrier to acid molecules after adsorbing copper, delaying the corrosion of copper in hydrochloric acid [78]. EC-AFM analysis confirmed that ZnO nanoparticles promoted the adsorption of ECDPA. Adsorption/deposition of a small number of inhibitor molecules was found when ZnO was not added. After the addition of ZnO, the number of inhibitor molecules on the surface was significantly increased, and the size of the inhibitor molecule became larger as the ZnO calcination temperature increased, as shown in Figure 34.

The corrosion resistance tests of ECDPA, EZ3 (ECDPA–ZnO at 300 °C), and EZ5 (ECDPA–ZnO at 500 °C) showed that the composite had better protection performance than ECDPA alone. The maximum inhibition efficiency of ECDPA was approximately 75%, while the composite could be further improved to 78% (EZ3) and 81% (EZ5). The improved corrosion inhibition performance of ECDPA–ZnO may have been related to the inclusion of ZnO nanoparticles, which promoted the adsorption of ECDPA over Cu.

Shaban et al. [79] studied the inhibition of dibenzylsulfoxide (DBSO) and *p*-chlorobenzohydroxamic acid (*p*-Cl-BHA) on copper corrosion in 0.5 M NaCl and 0.1 M Na_2_SO_4_, respectively. In situ AFM was used to observe the corrosion and inhibition process of the electrode surface. When DBSO was not added, the surface became rougher due to the dissolution of copper. In the presence of DBSO, the surface was not subject to severe corrosion and a relatively smooth surface was formed, as shown in Figure 35. Therefore, the authors thought that DBSO inhibited the corrosion of copper in sulfate solutions by converting it to a more stable, less soluble sulfide compound.

In the absence of *p*-Cl-BHA, the formation and growth of pitting corrosion were detected. The addition of *p*-Cl-BHA significantly impeded the localized corrosion of copper and inhibited the production of corrosion products that occurred when *p*-Cl-BHA was not added, as shown in Figure 36. Thus, the formation of a stable complex by adsorbing a corrosion inhibitor on a corroded surface effectively hindered further dissolution of the metal.

Cruickshank et al. [80] studied the anodic dissolution of polycrystalline copper in acid medium with or without corrosion inhibitor using EC-AFM. In 0.5 M H_2_SO_4_, the preferential corrosion of some grain surfaces occurred and then dissolved along grain boundaries. The addition of benzotriazole (BTA) formed a protective film that effectively inhibited the dissolution of copper. When the anode potential reached 200 mV, the protective film was stable. However, when the potential was as high as 300 mV, the film underwent local breakdown. In addition, Li et al. [81] investigated the effect of BTA on corrosion inhibition of copper using in situ EC-AFM in 0.01 M NaHCO_3_. The addition of BTA caused the copper surface to form a BTA film, effectively protecting the copper from erosion. The pitting potential of the copper surface covered by the BTA film could be significantly increased by more than 700 mV. Therefore, the author believed that, in fact, pitting was not a concern in the presence of BAT.

Through the morphology testing of EC-AFM, the adsorption–desorption states of corrosion inhibitors on an electrode surface can be intuitively understood, which is very beneficial for the exploration of electrochemical mechanisms of corrosion inhibitors. The application of EC-AFM in corrosion inhibitor protection is summarized in Table 7.

## 4. Conclusions and Perspectives

This review introduced the evolution, principles, and operation modes of EC-AFM, as well as illustrated its application in corrosion science through specific examples.

In summary, EC-AFM can not only perform real-time in situ research on micro-area corrosion (passivation) in the field of corrosion electrochemistry; it also has higher resolution, which provides detailed information on corrosion phenomena, such as activation and passivation at submicroscopic scales, especially in the early stages of corrosion. The surface topography changes of the target sample can be quantitatively analyzed with the information of surface height changes with applied voltage or time, helping us better study the corrosion process and mechanism. In recent years, EC-AFM was continuously improved in electrolytic cells, imaging modes, and probes. Moreover, EC-AFM was combined with SKPFM, SVET, and SECM to study the surface state of metals, greatly expanding the scope of application.

However, EC-AFM also has some problems that affect research, which brings about difficulties obtaining reliable surface topography features, such as drift during the scanning process, alignment of the laser, changes in the refractive index of the electrochemical medium as the process of corrosion goes on, finding accurate resonant frequencies, etc. In addition, the EC-AFM scanning rate is limited, and some rapid interface reactions cannot be monitored in real time.

In the future, the development of EC-AFM will be toward multi-functionality, high sensitivity, high speed, and high efficiency. The continuous improvement of EC-AFM will help carrying out more in-depth research on the in situ dynamic corrosion process, as well as promoting the study of corrosion science.

## Figures and Tables

**Figure 1 materials-13-00668-f001:**
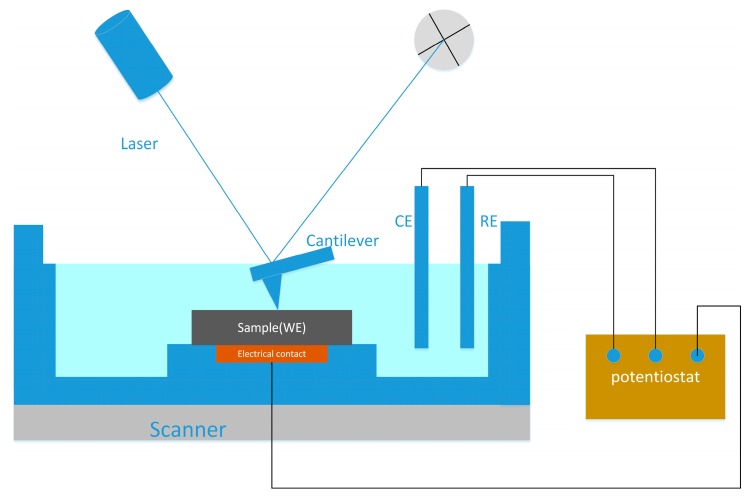
The schematic diagram of electrochemical atomic force microscopy (EC-AFM) electrolytic cell.

**Figure 2 materials-13-00668-f002:**
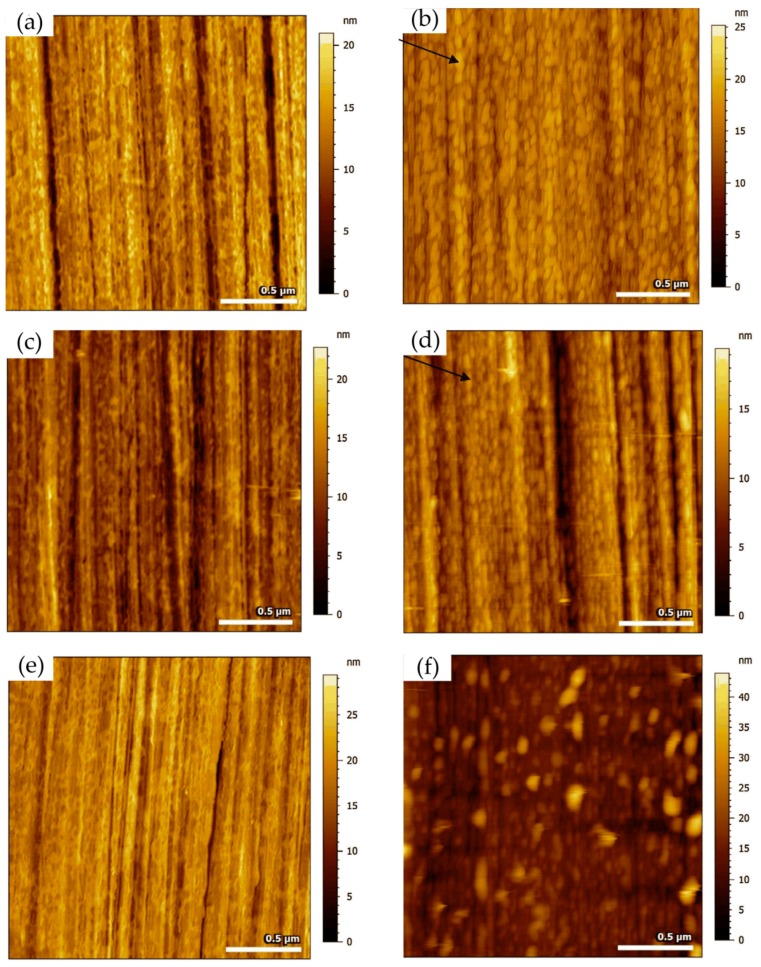
The AFM images of the steel specimen polarized at −0.1 V_(SCE__—saturated calomel electrode)_ for (**a**) 0.5 min and (**b**) 60 min, at 0.5 V_(SCE)_ for (**c**) 0.5 min and (**d**) 60 min, and at 0.7 V_(SCE)_ for (**e**) 0.5 min and (**f**) 60 min. The black arrow points to scale-like spots. Adapted with permission from Reference [33]; copyright 2017 Elsevier Ltd.

**Figure 3 materials-13-00668-f003:**
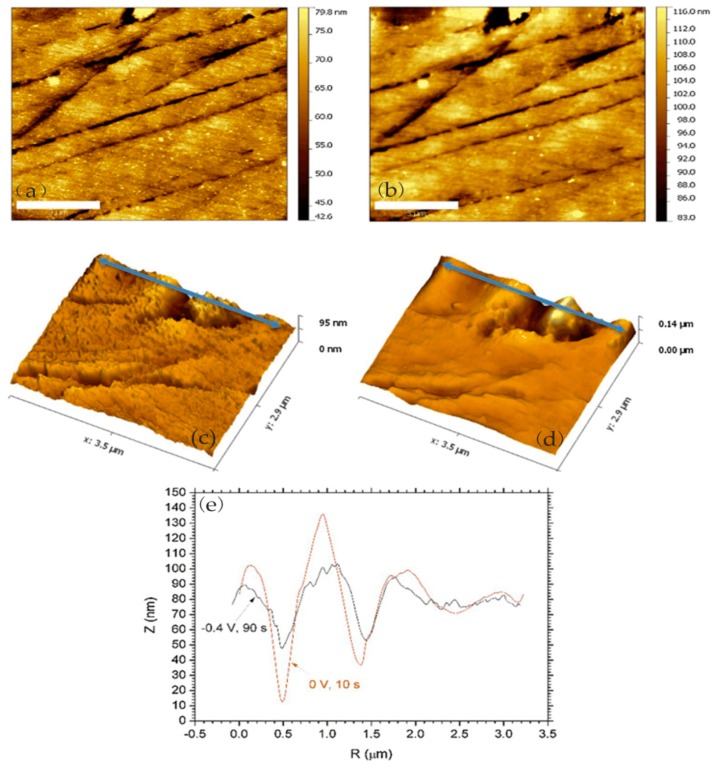
The in situ AFM images of AA 2024: (**a**) 226 min, −0.4 V/Ag/AgCl/KCl_sat_, 90 s; (**b**) 328 min, 0 V/Ag/AgCl/KCl_sat_, 10 s. (**c**,**d**) The corroding sites on top of the images (**a**,**b**) as three-dimensional (3D) images. (**e**) The profile lines measured over the corroding intermetallic particle. Reprinted with permission from Reference [35]; copyright 2016 Elsevier Ltd.

**Figure 4 materials-13-00668-f004:**
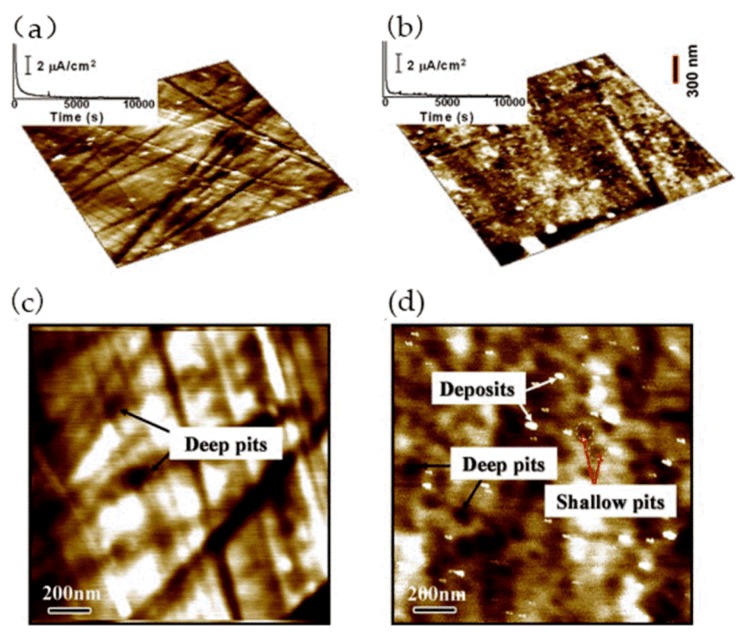
AFM imagines of the surfaces of amorphous and crystallized Ni50Nb50 alloys after polarization at 1.0 V in 1 mol/L HCl solution. (**a**,**b**) Surface morphologies of the amorphous and crystallized samples. Inserts are the corresponding I–t curves during potentiostatic treatments. (**c**,**d**) High-magnification images of selective areas in amorphous and crystallized samples. Adapted with permission from Reference [39]; copyright 2010 Elsevier Ltd.

**Figure 5 materials-13-00668-f005:**
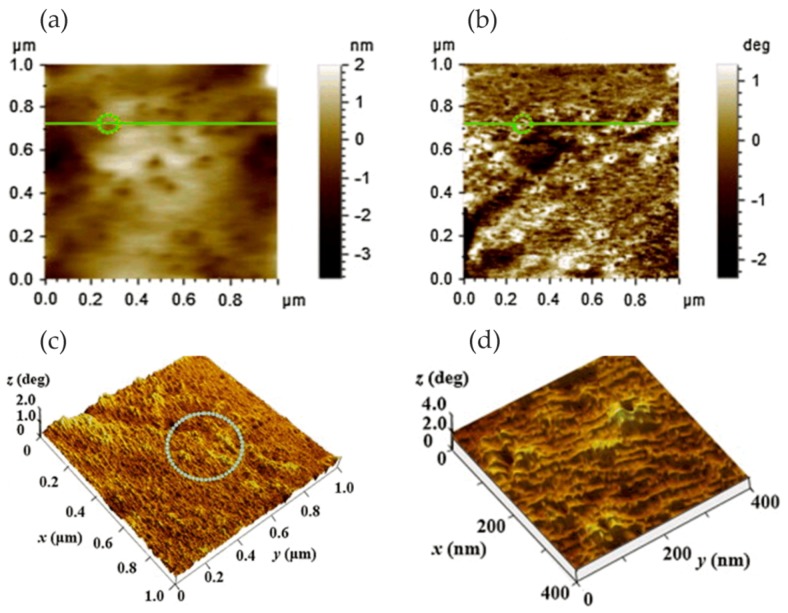
In situ tapping mode (TM) AFM images of the annealed Al_88_Ni_8_Ce_4_ amorphous nanocrystalline sample at open-circuit potential (OCP) held for 3 min in 0.01 mol/L NaCl solution. Topography image (**a**) and phase image (**b**). Three-dimensional (3D) images of phase (**c**) and the (**d**) local enlarged phase image of the marked circle in (**c**). Reprinted with permission from Reference [40]; copyright 2014 Elsevier Ltd.

**Figure 6 materials-13-00668-f006:**
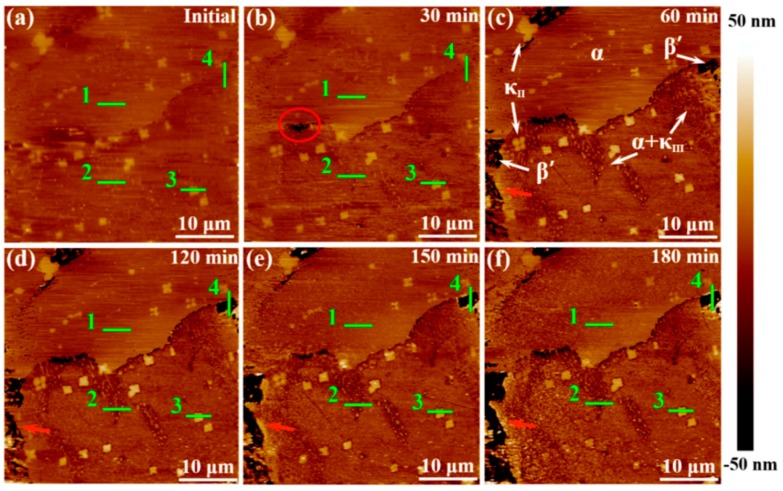
In situ topography images of nickel–aluminum bronze (NAB) specimen surface after exposure to 3.5 wt% NaCl solution at different times: (**a**) initial; (**b**) 30 min; (**c**) 60 min; (**d**) 120 min; (**e**) 150 min; (**f**) 180 min. Site 1 corresponds to the α phase, site 2 corresponds to the α + κ_III_ eutectoid structure, and site 3 corresponds to the κ_II_ phase, while site 4 corresponds to the β’ phase. Reprinted with permission from Reference [44]; copyright 2019 MDPI.

**Figure 7 materials-13-00668-f007:**
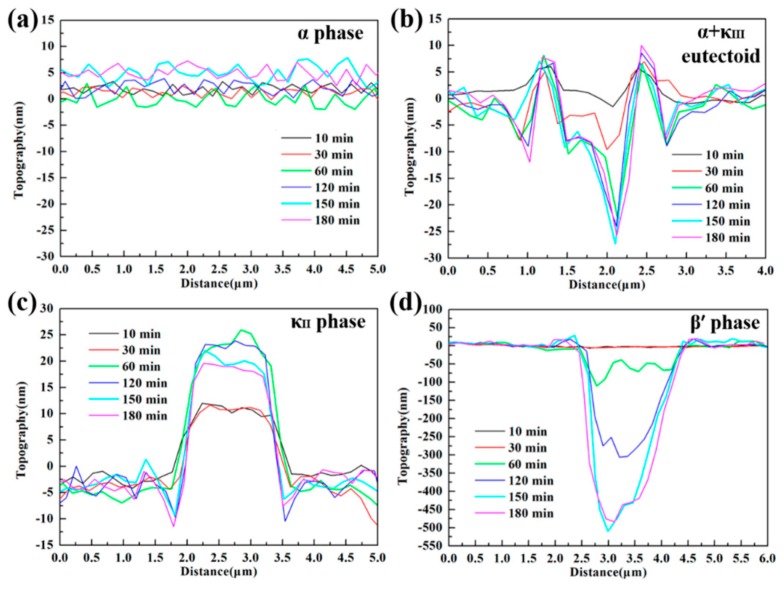
In situ line profiles corresponding to sites 1, 2, 3, and 4 marked in Figure 6, respectively: (**a**) α phase; (**b**) α + κ_III_ eutectoid structure; (**c**) κ_II_ phase; (**d**) β’ phase. Reprinted with permission from Reference [44]; copyright 2019 MDPI.

**Figure 8 materials-13-00668-f008:**
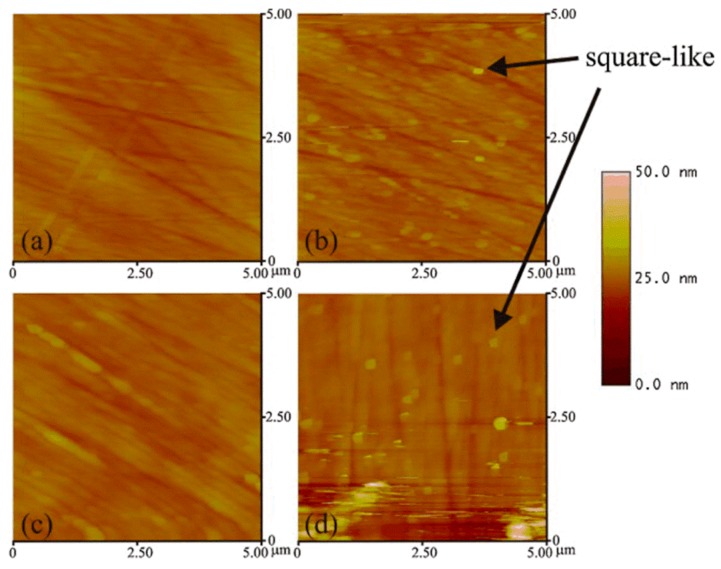
AFM images of DSS 2205 sample: (**a**) in artificial saliva (AS) after being exposed to the potential of 0.8 V for 45 min, (**b**) in AS after additional exposure to the potential of 1 V for 11.6 min, (**c**) in physiological solution (PS) after being exposed to a potential of 0.8 V for 17.2 min, and (**d**) in PS after additional exposure to 1 V for 7.5 min. Adapted with permission from Reference [45]; copyright 2011 Elsevier Ltd.

**Figure 9 materials-13-00668-f009:**
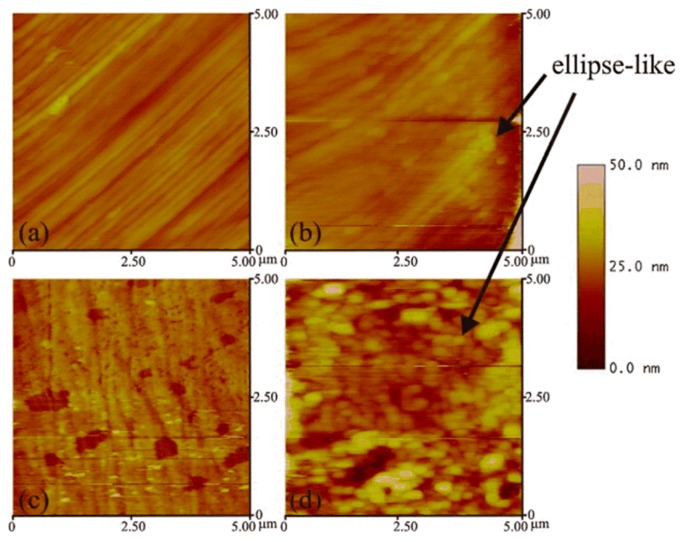
AFM image of AISI 316L sample: (**a**) in AS after exposing the sample to anodic potential of 0.5 V for 10 min, (**b**) in AS after additional exposure of the sample to 0.5 V for 6.6 min, (**c**) in PS immediately after the test cyclic voltammogram in the range of potentials from −0.5 V to 0.8 V, and (**d**) in PS after additional exposure to 0.5 V for 30 s. Adapted with permission from Reference [45]; copyright 2011 Elsevier Ltd.

**Figure 10 materials-13-00668-f010:**
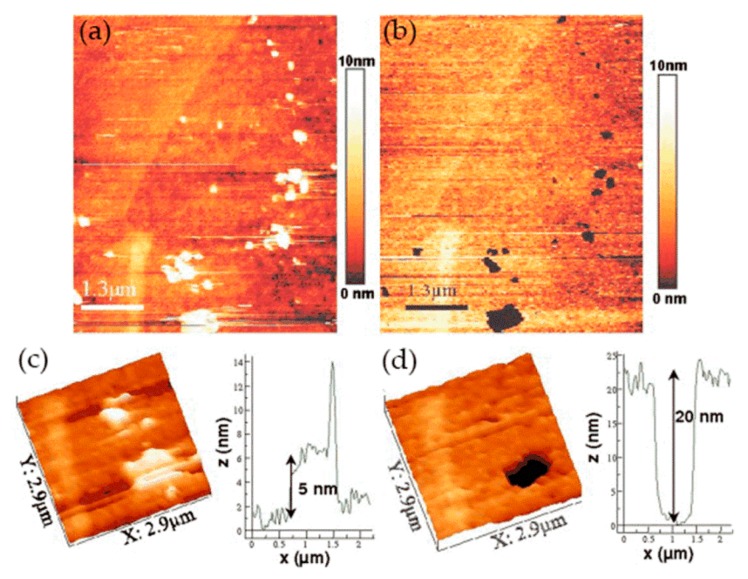
AFM images of the surface at open-circuit potential (**a**) and at pitting potential (**b**). (**c**,**d**) The pit chain (in black, right image) profiles made on images at higher magnification (bottom insets) show that the average relief of the islands was about 5 nm, whereas the average depth of the pits was about 20 nm. Reprinted with permission from Reference [46]; copyright 2008 Elsevier Ltd.

**Figure 11 materials-13-00668-f011:**
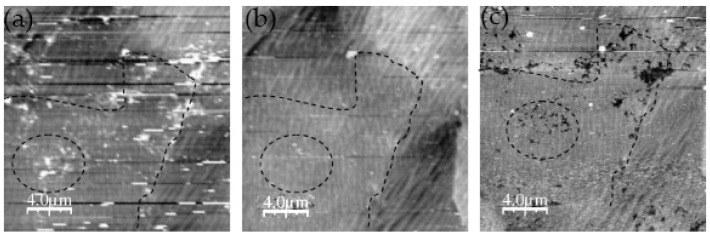
AFM images of the surface, taken at (**a**) open-circuit potential, (**b**) corrosion potential after cathodic scan, and (**c**) pitting potential. Reprinted with permission from Reference [46]; copyright 2008 Elsevier Ltd.

**Figure 12 materials-13-00668-f012:**
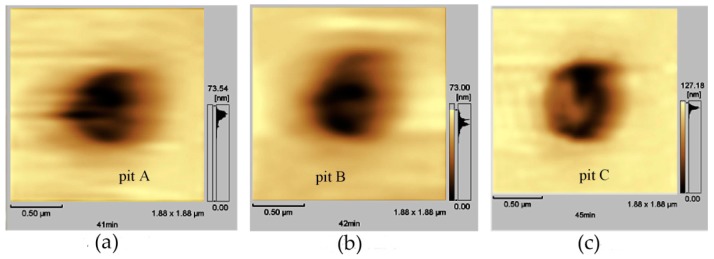
AFM images of solution-treated SUS304 stainless steel during corrosion in 3.5 wt% sodium chloride solution at 298 K (I = 10 A/m^2^): (**a**) t = 2.46 ks; (**b**) t = 2.52 ks; (**c**) t = 2.7 ks. Reprinted with permission from Reference [48]; copyright 2005 Elsevier Ltd.

**Figure 13 materials-13-00668-f013:**
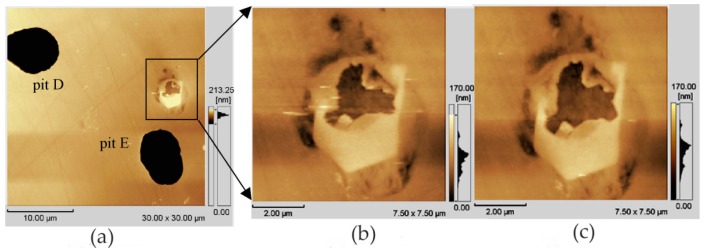
AFM images of solution-treated SUS304 stainless steel after corrosion in sodium chloride solution at 298 K (I = 5 A/m^2^): (**a**) t = 1.5 ks; (**b**) t = 1.5 ks and (**c**) t = 7.2 ks are the magnified images of the framed area of (**a**), showing the presence of the corrosion product. Reprinted with permission from Reference [48]; copyright 2005 Elsevier Ltd.

**Figure 14 materials-13-00668-f014:**
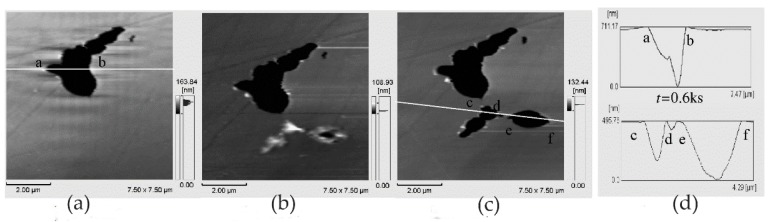
AFM images of sensitization-treated SUS304 stainless steel after corrosion in sodium chloride solution (I = 1 A/m^2^): (**a**) t = 0.6 ks; (**b**) t = 0.9 ks; (**c**) t = 1.2 ks. (**d**) Cross-sectional profiles of pits along a–b and c–f lines. Reprinted with permission from Reference [48]; copyright 2005 Elsevier Ltd.

**Figure 15 materials-13-00668-f015:**
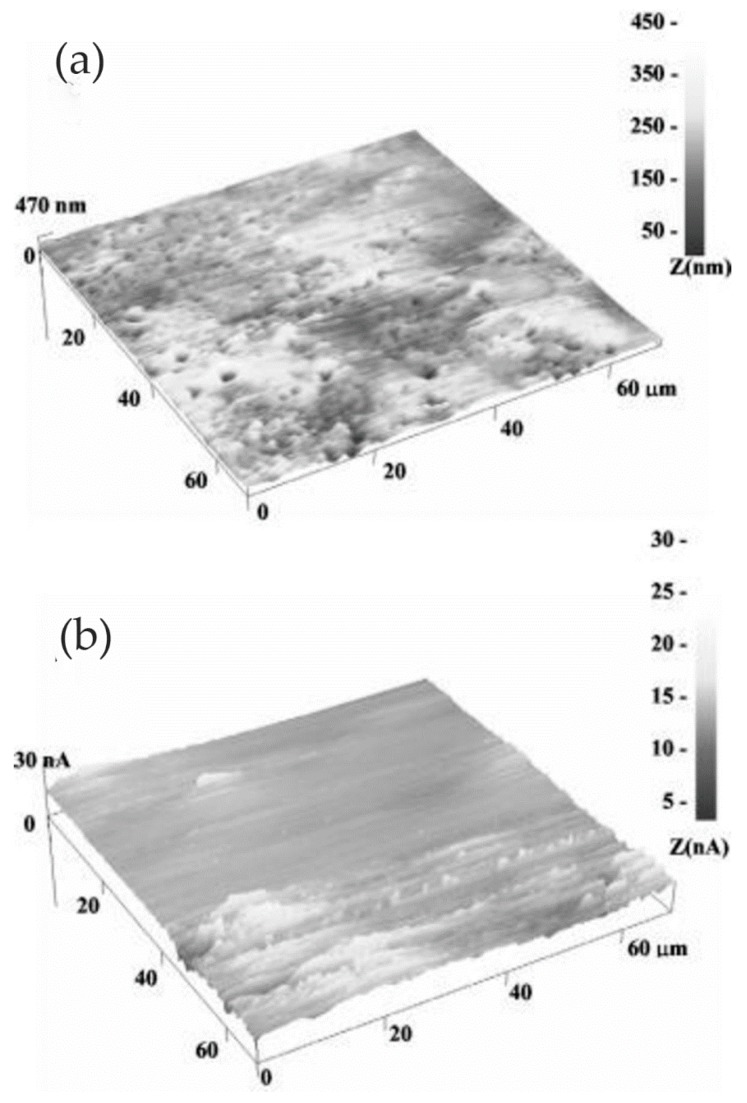
EC-AFM/scanning electrochemical microscopy (SECM) images of AA1050 in 10 mM NaCl+ 5 mM KI at 300 mV anodic polarization, and tip at +750 mV vs. Ag/AgCl: (**a**) topography; (**b**) electrochemical activity map. Adapted with permission from Reference [50]; copyright 2005 Electrochemical Society.

**Figure 16 materials-13-00668-f016:**
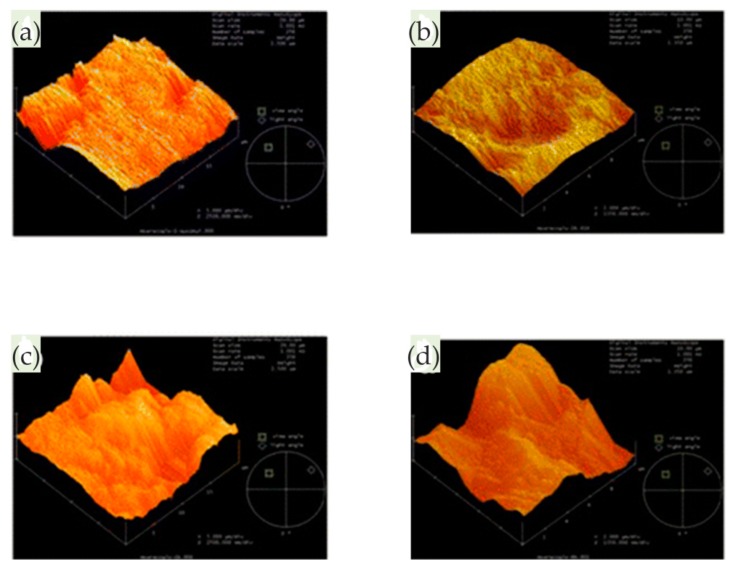
AFM images recorded for Zn in 0.5 M NaOH solution containing 0.05 M (**a**,**b**) ClO_4_^−^ or (**c**,**d**) ClO_3_^−^. Adapted with permission from Reference [51]; copyright 2013 Springer.

**Figure 17 materials-13-00668-f017:**
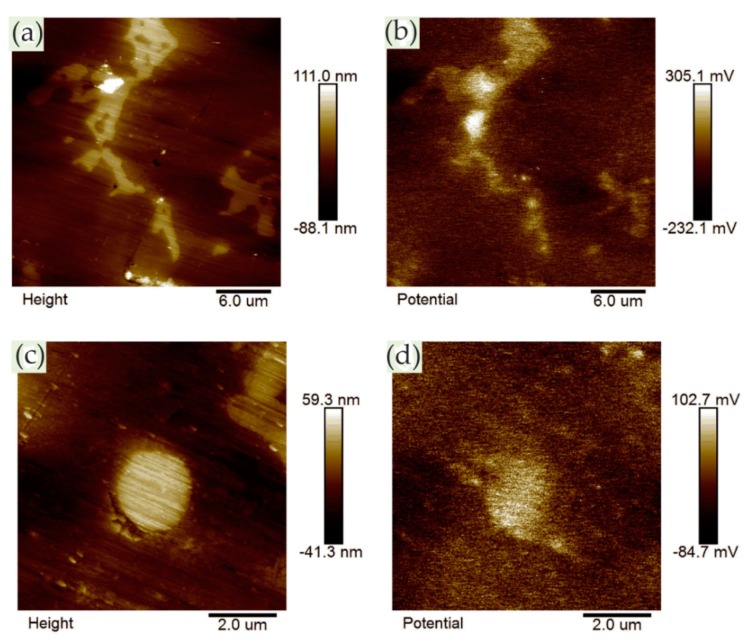
The polished AZ91 alloy (**a**) surface topography and (**b**) relative Volta potential map of elongated b-Mg_17_Al_12_; (**c**) surface topography and (**d**) relative Volta potential map of granular-like b-Mg_17_Al_12_. Reprinted with permission from Reference [53]; copyright 2019 Elsevier Ltd.

**Figure 18 materials-13-00668-f018:**
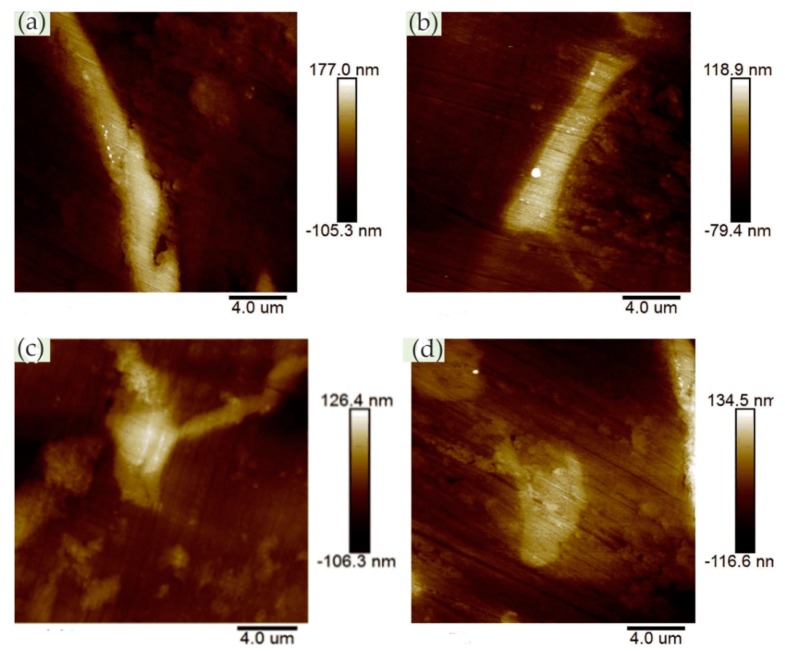
Typical in situ surface images for different intermetallics after immersion in 0.1 mol/L NaCl solution for 60 min: (**a**) acicular-like RE phase, (**b**) rod-like RE phase, (**c**) granular RE phase, and (**d**) dispersed granular b-Mg17Al12 in AZ91 alloy with 1.0% (La,Ce) mischmetal (MM) addition. Reprinted with permission from Reference [52]; copyright 2019 Elsevier Ltd.

**Figure 19 materials-13-00668-f019:**
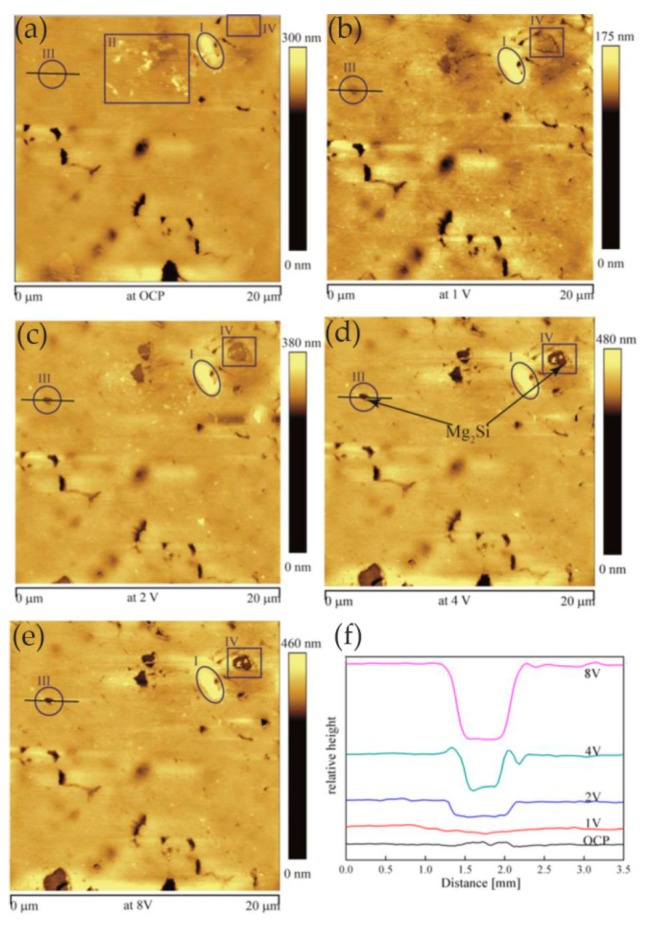
In situ EC-AFM topography images of the Al 6060 alloy in 0.2 M Na_2_SO_4_ solution at (**a**) OCP, (**b**) 1 V, (**c**) 2 V, (**d**) 4 V, and (**e**) 8 V vs. Ag/AgCl. (**f**) Line profiles across the feature III in (**a**–**d**). Adapted with permission from Reference [54]; copyright 2016 Electrochemical Society.

**Figure 20 materials-13-00668-f020:**
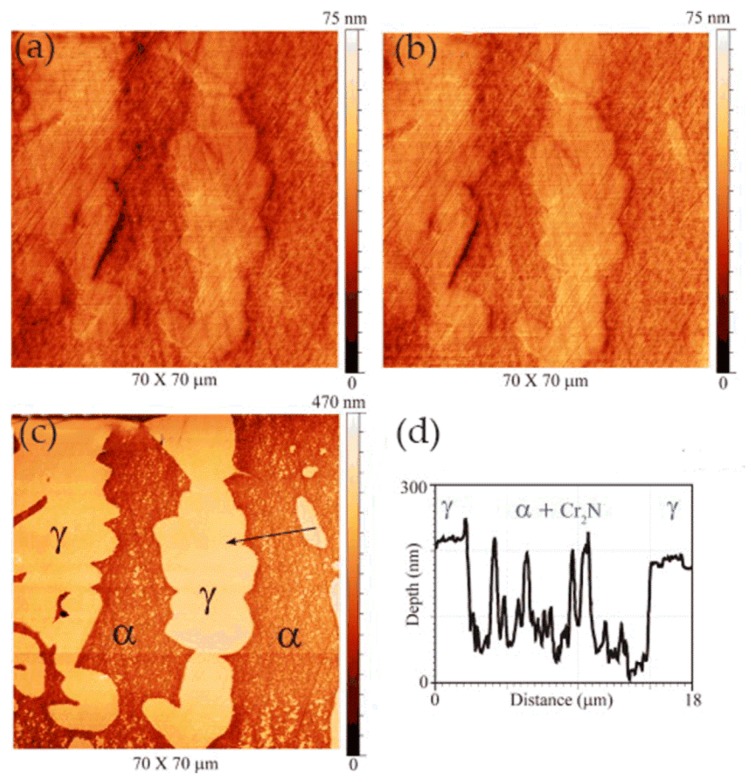
In situ EC-AFM images of the same area of the 2205 HT in 1 M NaCl at room temperature under electrochemical control at different applied potentials: (**a**) OCP, (**b**) 1.1 V_Ag/AgCl_, and (**c**) 1.2 V_Ag/AgCl_. (**d**) Depth line profile from the arrow in image (**c**) showing particles or particle clusters remaining in the dissolved ferrite area. Reprinted with permission from Reference [55]; copyright 2013 Elsevier Ltd.

**Figure 21 materials-13-00668-f021:**
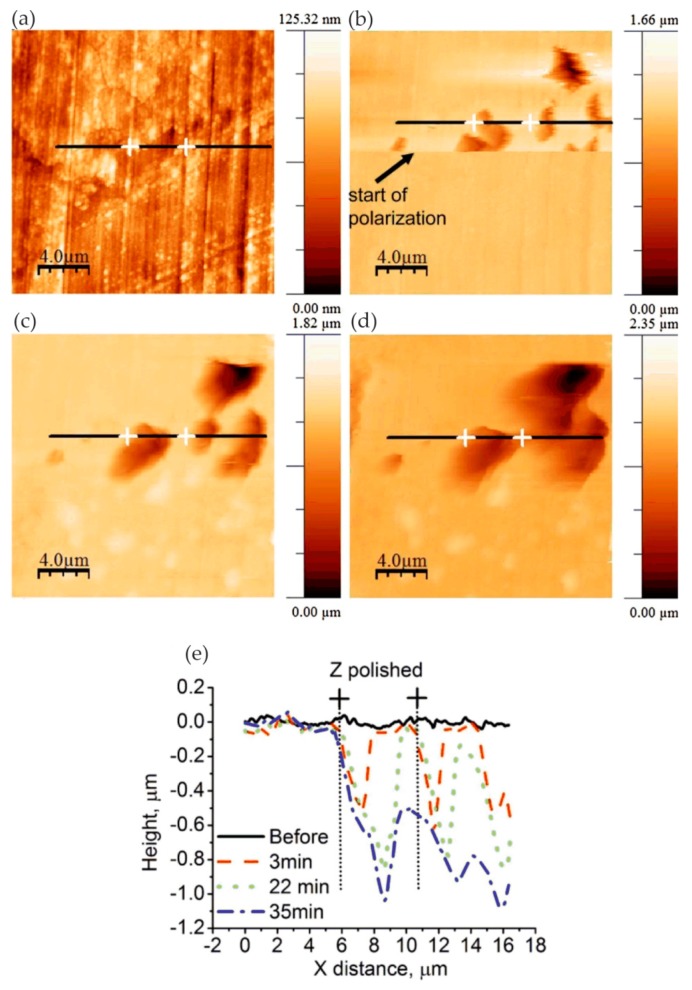
AFM topography maps made on cutting edge of adhesively bonded Zn (Z) substrate before immersion (**a**) and during 24 min (**b**), 1 h 5 min (**c**), and 2 h (**d**) of immersion in 0.001 M NaCl. (**e**) Evolution of topography across the black line profiles drawn in the same zones on AFM maps. Black arrows indicate local pits on the zinc layer. Reprinted with permission from Reference [56]; copyright 2016 Elsevier Ltd.

**Figure 22 materials-13-00668-f022:**
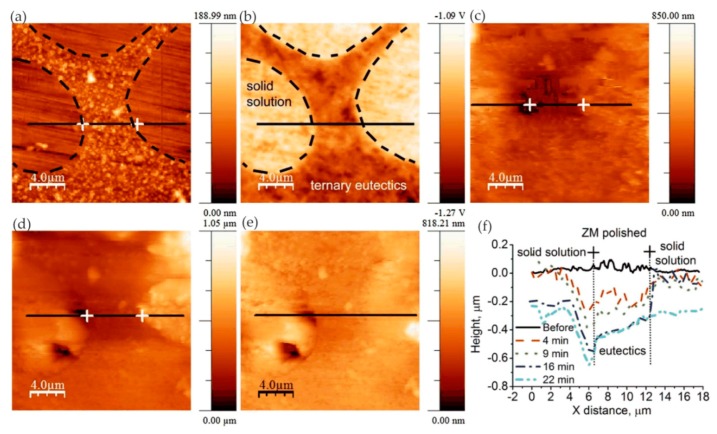
AFM topography (**a**) and SKPFM (**b**) maps of Zn–Al–Mg (ZM) galvanized coating before immersion and during immersion in 0.005 M NaCl for 4 min (**c**), 9 min (**d**), and 22 min (**e**). (**f**) Evolution of topography across the black line profiles. Reprinted with permission from Reference [56]; copyright 2016 Elsevier Ltd.

**Figure 23 materials-13-00668-f023:**
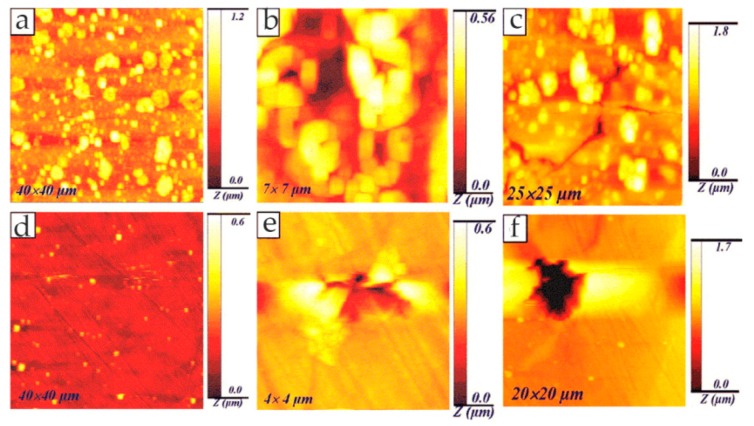
In situ AFM images of (**a**) EN AW-3003 and (**b**) Al–Mn–Si–Zr, after two days; (**e**) EN AW-3003 and (**f**) Al–Mn–Si–Zr, after 3.5 days in Sea Water Acetic Acid Test (SWAAT) solution of pH 4. (**c**,**d**) Magnified images of the framed areas of (**a**,**b**), respectively. Adapted with permission from Reference [58]; copyright 2007 Elsevier Ltd.

**Figure 24 materials-13-00668-f024:**
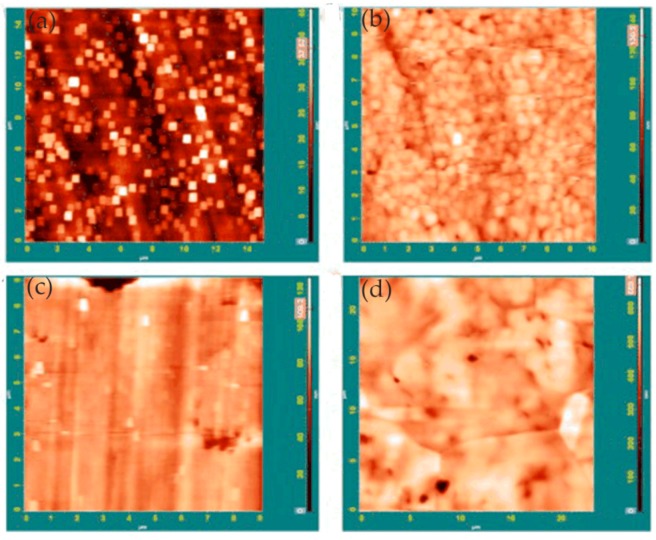
Surface morphology of 304L stainless steel at 1100 mV in (**a**) 0.1 M HNO_3_, (**b**) 0.5 M HNO_3_, (**c**) 0.6 M HNO_3_, and (**d**) 1 M HNO_3_. Reprinted with permission from Reference [60]; copyright 2010 Elsevier Ltd.

**Figure 25 materials-13-00668-f025:**
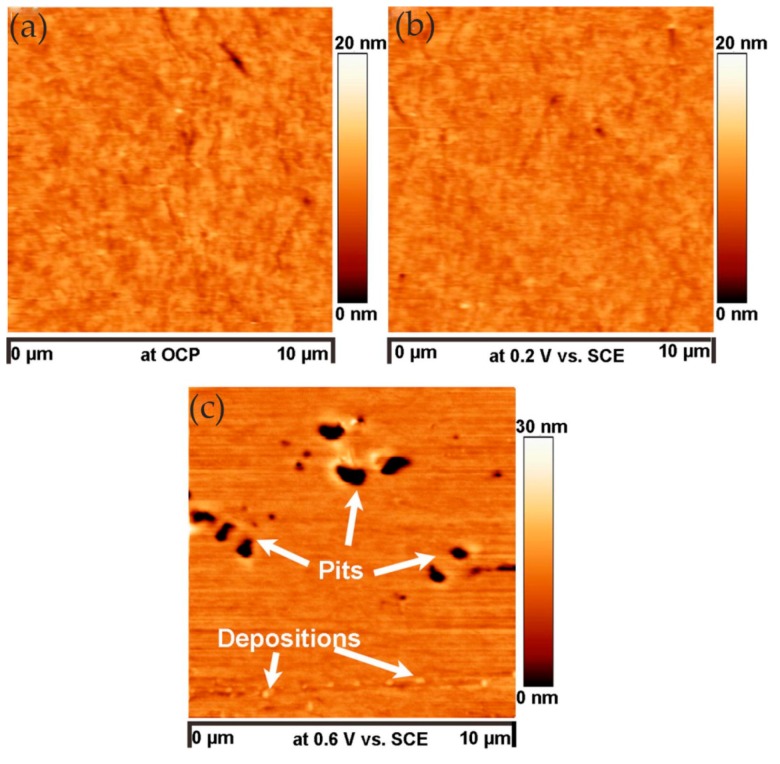
In situ EC-AFM topography images of the Al_0.3_CoCrFeNi high-entropy alloy (HEA) in a 3.5 wt% NaCl solution after 10 min of exposure at (**a**) OCP, (**b**) 0.2 V, and (**c**) 0.6 V. Reprinted with permission from Reference [63]; copyright 2018 Elsevier Ltd.

**Figure 26 materials-13-00668-f026:**
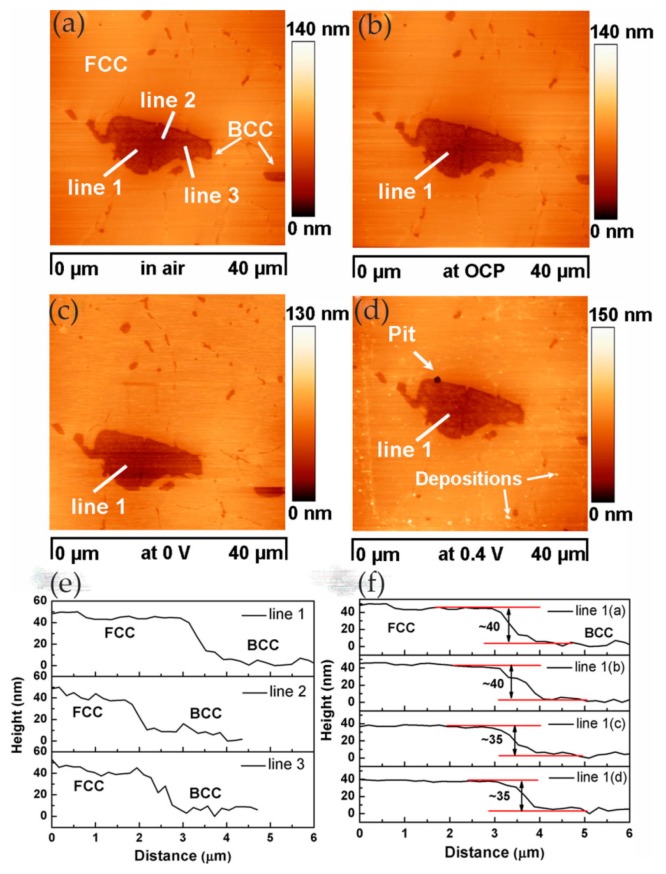
The AFM topography image in air (**a**), and in situ EC-AFM topography images of the Al_0.5_CoCrFeNi HEA after 10 min of exposure in a 3.5 wt. % NaCl solution at (**b**) OCP, (**c**) 0 V, and (**d**) 0.4 V. (**e**) Three-line profiles across the phase boundaries illustrated in (**a**). (**f**) Vertical profiles along line 1 of the surface in (**a**–**d**). Reprinted with permission from Reference [63]; copyright 2018 Elsevier Ltd.

**Figure 27 materials-13-00668-f027:**
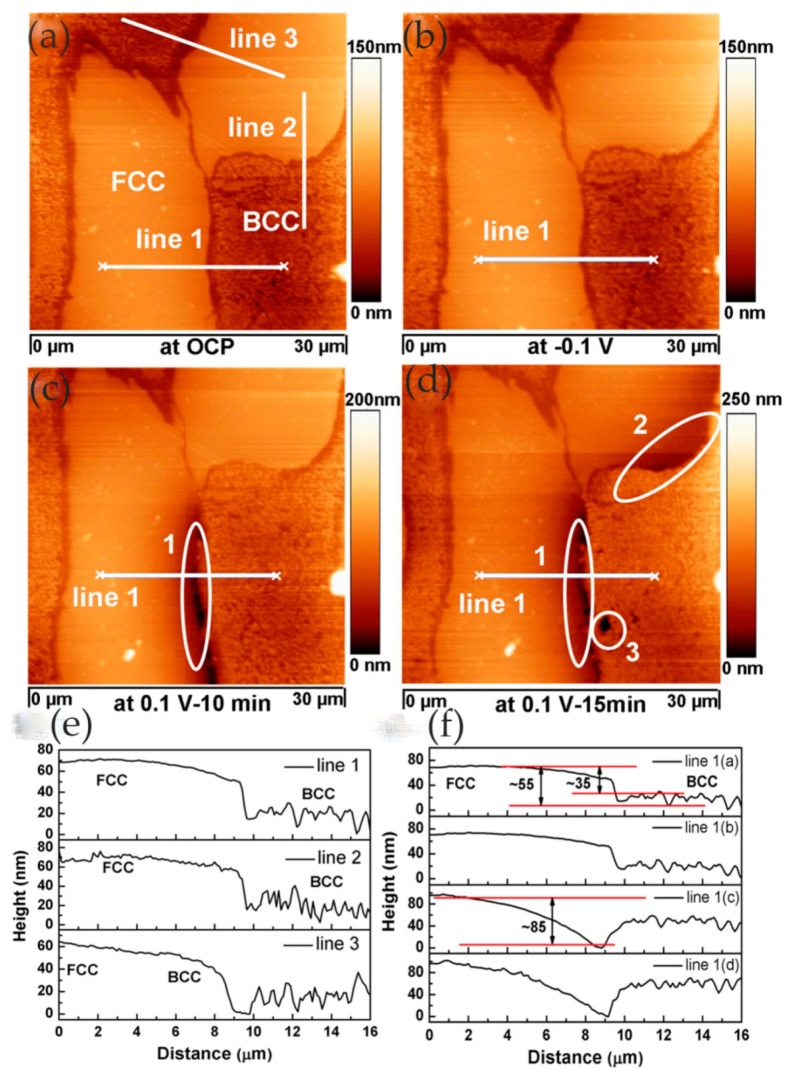
In situ EC-AFM topography images of the Al_0.7_CoCrFeNi HEA after 10 min of exposure in a 3.5 wt% NaCl solution at (**a**) OCP, (**b**) 0.1 V, and (**c**) 0.1 V, and (**d**) after 15 min of exposure at 0.1 V vs. SCE. (**e**) Three-line profiles across the phase boundaries illustrated in (**a**). (**f**) Vertical profiles along line 1 of the surface in (**a**–**d**). Reprinted with permission from Reference [63]; copyright 2018 Elsevier Ltd.

**Figure 28 materials-13-00668-f028:**
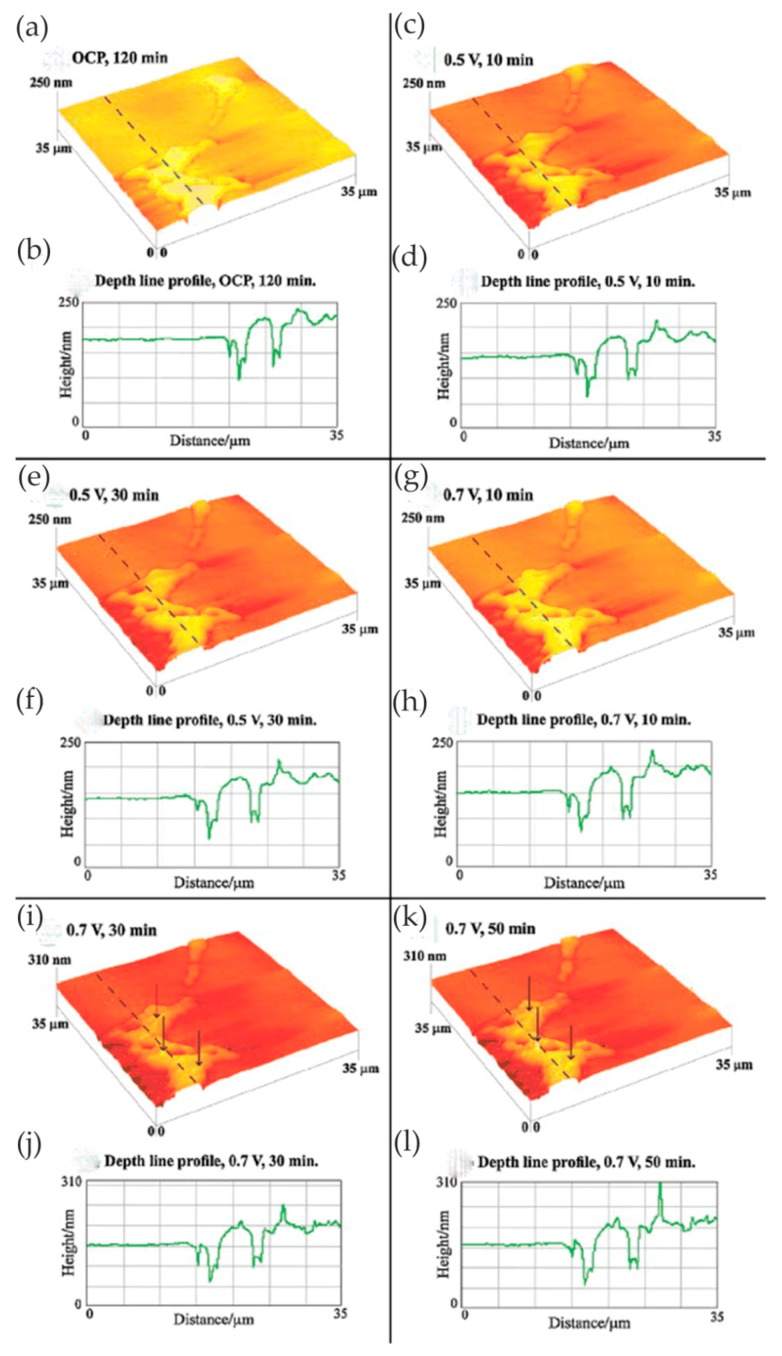
In situ AFM images of the same area of CoCrMo alloy under electrochemical control in PBS solution with pH 7.4, (**a**) after 120 min at OCP, (**c**,**e**) after 10 and 30 min at 0.5 Vsat Ag/AgCl, and (**g**,**i**,**k**) after 10, 30, and 50 min at 0.7 Vsat Ag/AgCl, with marked etching-like dissolution sites on the carbide (where visible). (**b**,**d**,**f**,**h**,**j**,**l**) Depth line profiles at the applied potential and time. Adapted with permission from Reference [64]; copyright 2011 Elsevier Ltd.

**Figure 29 materials-13-00668-f029:**
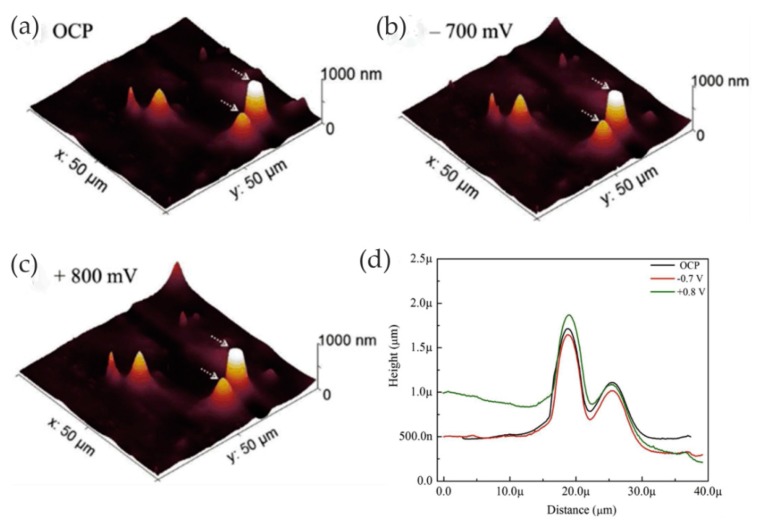
EC-AFM images (3D) of the same scan area of the composite coating in 3.0 wt% NaCl solution obtained after 60 min (**a**) at OCP, (**b**) at −700 mV (vs. Ag/AgCl), and (**c**) at 800 mV (vs. Ag/AgCl). (**d**) Line profiles of CeO_2_ nanoparticle (CeNP) aggregates when at the two applied potentials drawn by the crossing line. Reprinted with permission from Reference [66]; copyright 2019 Elsevier Ltd.

**Figure 30 materials-13-00668-f030:**
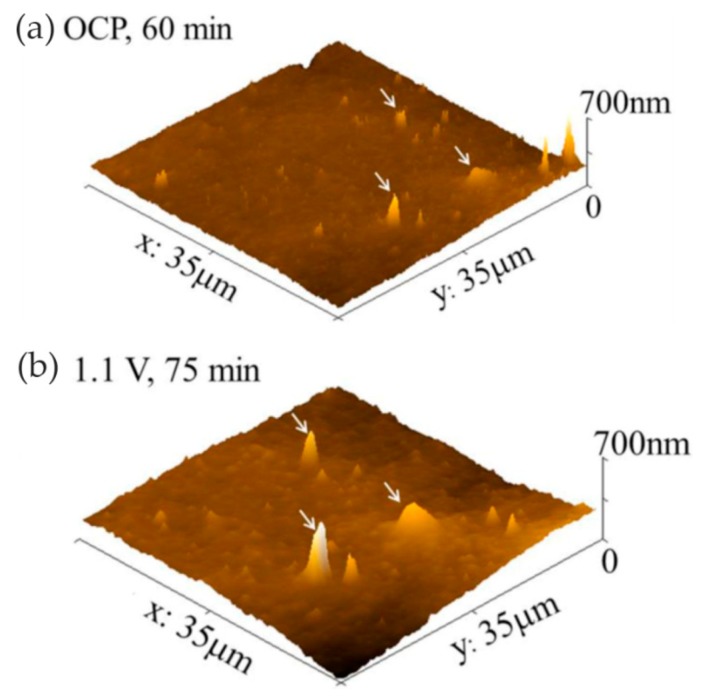
EC-AFM images (3D) of the same area of the composite coating obtained after 14 h of exposure in 3.0 wt% NaCl solution: (**a**) after 60 min at OCP; (**b**) after 75 min at 1.1 V vs. Ag/AgCl. Adapted with permission from Reference [68]; copyright 2015 Electrochemical Society.

**Figure 31 materials-13-00668-f031:**
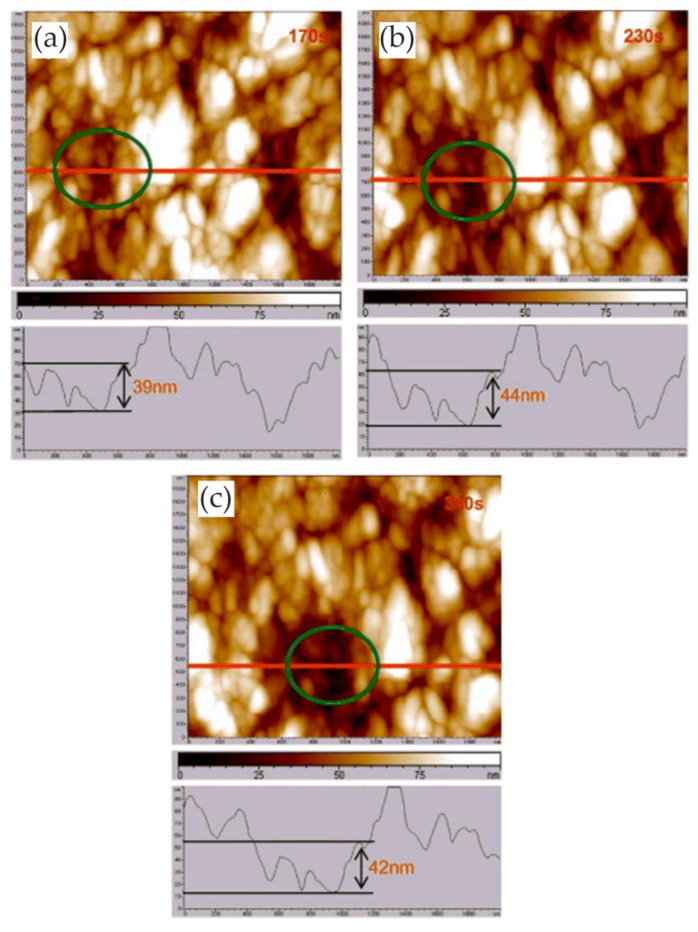
In situ AFM images of nanocrystalline (NC) thin film in the initial pitting stage under anodic polarization in 3.5 wt% NaCl solution: (**a**) 170 s; (**b**) 230 s; (**c**) 340 s. Reprinted with permission from Reference [71]; copyright 2013 Elsevier Ltd.

**Figure 32 materials-13-00668-f032:**
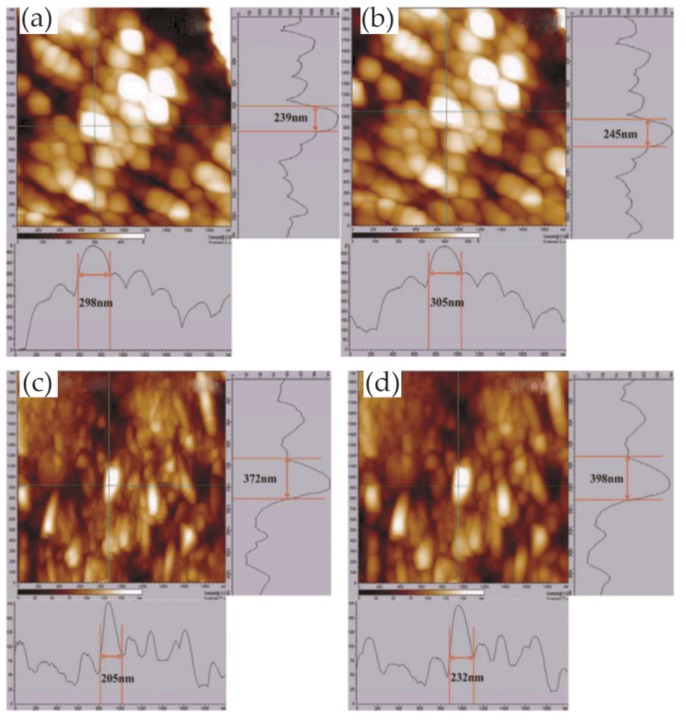
In situ AFM images of coarse crystal (CC) 304L stainless steel (**a**,**b**) and a NC thin film (**c**,**d**) (scale 2 μm × 2 μm) in the growth stage of a passive film under anodic polarization in 0.05 M H_2_SO_4_ + 0.2 M NaCl solution after passivation of 6 min (**a**,**c**) and 12 min (**b**,**d**). Reprinted with permission from Reference [72]; copyright 2012 Electrochemical Society.

**Figure 33 materials-13-00668-f033:**
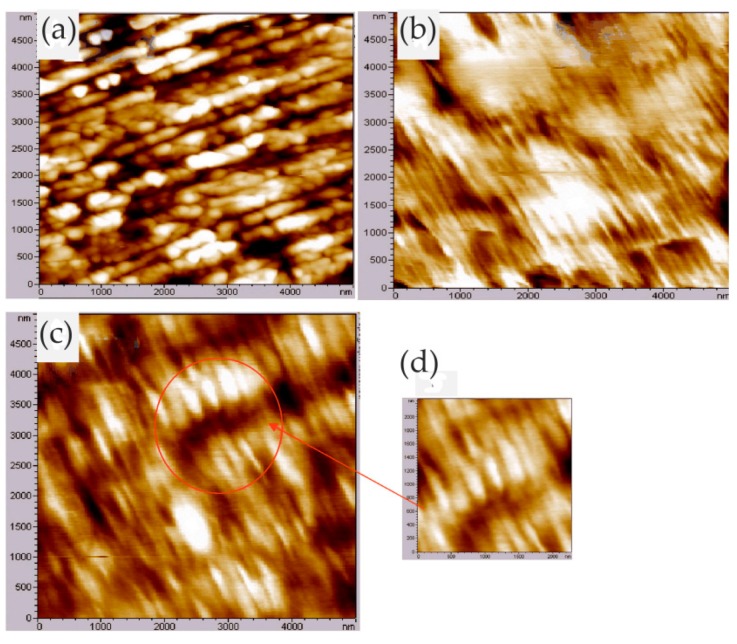
In situ AFM image of Al in 0.5 M H_2_SO_4_ in the presence of polyacrylic acid (PAA) + KI at different potentials: (**a**) −1.0 V, (**b**) −0.70 V, and (**c**) −0.50 V. (**d**) Magnified images of the framed area of (**c**). Adapted with permission from Reference [76]; copyright 2013 Taylor & Francis.

**Figure 34 materials-13-00668-f034:**
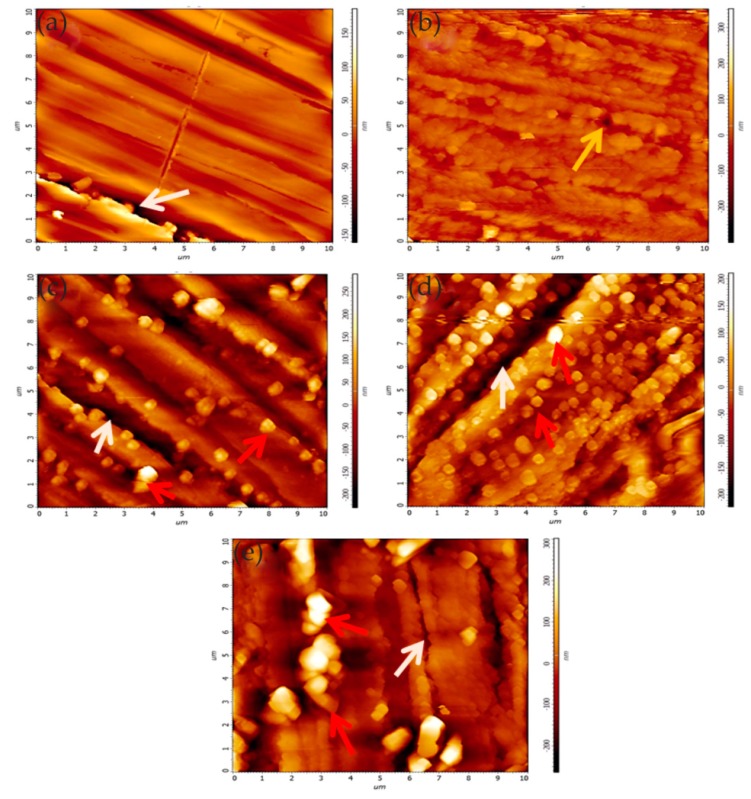
EC-AFM two-dimensional (2D) images of (**a**) polished copper, and copper (**b**) corroded, as well as inhibited by (**c**) ethyl-2-cyano-3-(4-(dimethylamino) phenyl) acrylate (ECDPA), (**d**) EZ3 (ECDPA–ZnO at 300 °C), and (**e**) EZ5 (ECDPA–ZnO at 500 °C) in 10 min of immersion in 1 M HCl at −0.005 V. Reprinted with permission from Reference [78]; copyright 2019 Elsevier Ltd.

**Figure 35 materials-13-00668-f035:**
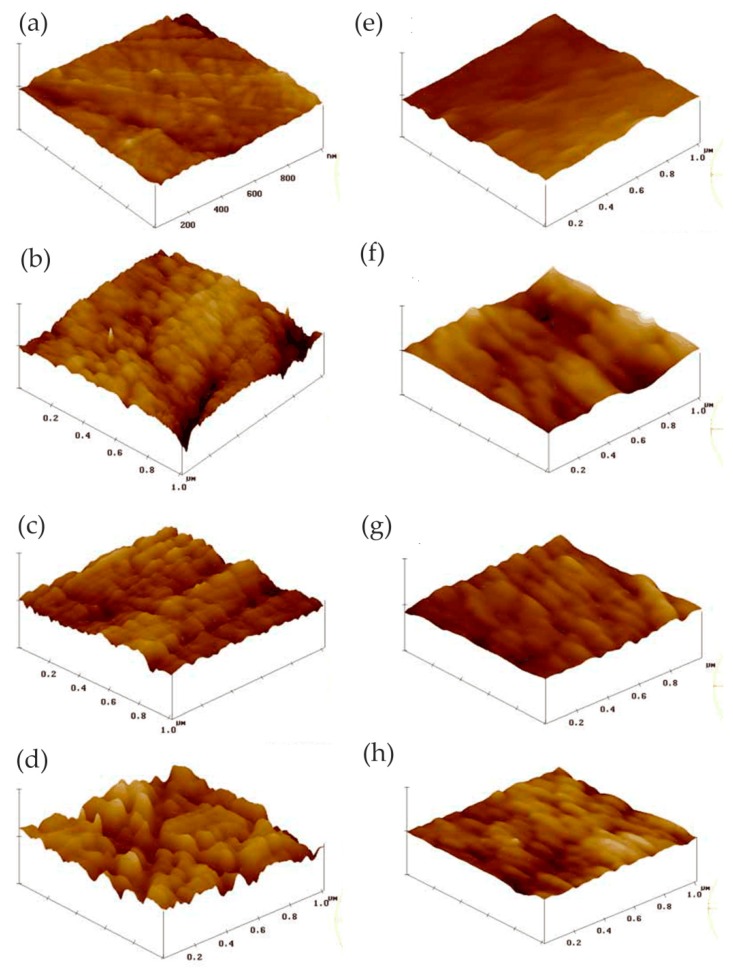
Morphological changes of copper in solution of 0.1 M Na_2_SO_4_ (**a**–**d**) and 0.1 M Na_2_SO_4_ + 0.5 mM dibenzylsulfoxide (DBSO) (**e**–**h**), at different times: (**a**,**e**) 0 min; (**b**,**f**) 15 min; (**c**,**g**) 30 min; (**d**,**h**) 45 min. Reprinted with permission from Reference [79]; copyright 1998 Springer.

**Figure 36 materials-13-00668-f036:**
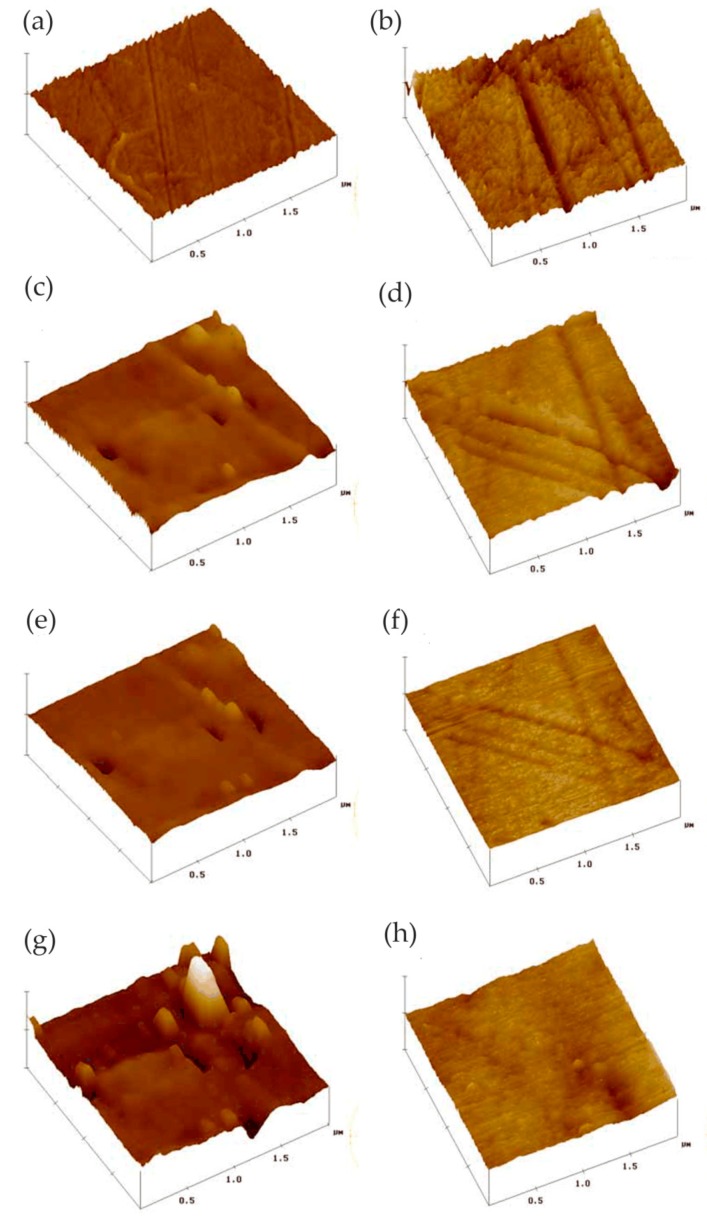
Morphological changes of copper in solution of 0.1 M NaCl (**a**–**d**) and 0.1 M NaCl + *p*-chlorobenzohydroxamic acid (*p*-Cl-BHA) (**e**–**h**), at different times: (**a**,**e**) 0 min; (**b**,**f**) 15 min; (**c**,**g**) 30 min; (**d**,**h**) 45 min. Reprinted with permission from Reference [79]; copyright 1998 Springer.

**Table 1 materials-13-00668-t001:** Comparison of common operation modes of EC-AFM.

Mode	Probes (Nominal Spring Constants)	Cantilever	Tip–Sample Distance	Force	Scan Rate
Contact mode	Silicon nitride probes (less than tapping and non-contact mode)	The cantilever is deflected: constant force or constant height	<0.5 nm	Repulsive 10^−9^–10^−6^ N	High scan speeds
Non-contact mode	Silicon probes (20–100 N/m)	The cantilever is oscillated (amplitude < 10 nm)	1–10 nm	Attractive 10^−12^ N	Slower scan speed than tapping and contact mode
Intermittent tapping (tapping mode)	Silicon probes (2–50 N/m)	The cantilever is oscillated (amplitude > 20 nm)	0.5–2 nm (intermittent contact)	Both repulsive and attractive forces	slower scan speed

**Table 2 materials-13-00668-t002:** Summary of the application of EC-AFM in corrosion product films.

	Materials	Solution	Mode	Reference
Corrosion product film	carbon steel	Carbonate/bicarbonate solution	Contact mode	[33]
	304L stainless steel	Nitric acid		[34]
	AA 2024-T3	0.5 M NaCl solution	Tapping mode	[35]
	CoCrMo alloy	Phosphate buffer saline (PBS) and simulated inflammation (SI) solution	Contact mode	[36]
	Ti-6Al-4V	PBS	Contact mode	[37]
	Ni50Nb50 metallic glass	1 mol/L HCl solution	Tapping mode	[39]
	Al–Ni–Ce metallic glass	0.01 mol/L NaCl solution	Tapping mode	[40]
	X100 pipeline steel	0.01 M NaHCO_3_ solution	Contact mode	[41]
	NAB	3.5 wt% NaCl solution	Contact mode	[44]

**Table 3 materials-13-00668-t003:** Summary of the application of EC-AFM in pitting corrosion.

	Materials	Solution	Mode	Reference
Pitting corrosion	AISI 316L and DSS 2205	Physiological solution and artificial saliva	Contact mode	[45]
	304L stainless steel	Chloride borate buffer solution	Contact mode	[46]
	SUS304	3.5 wt% NaCl solution		[48]
	pure aluminum	0.01 mol/L FeC1_3_ solution	Contact mode	[49]
	AA1050	10 mM NaCl + 5 mM KI	Contact mode	[50]
	Metallic zinc	0.5 M NaOH solution containing different concentrations of ClO_3_^−^ or ClO_4_^−^ anions		[51]
	Hot-dip-galvanized surface	0.01 M NaOH	Contact mode	[52]

**Table 4 materials-13-00668-t004:** Summary of the application of EC-AFM in selective corrosion.

	Materials	Solution	Mode	Reference
Selective corrosion	AZ91 alloy added (La,Ce) mischmetal	0.1 mol/L NaCl solution	Contact mode	[53]
	Al 6060 alloy	0.2 M Na_2_SO_4_ solution	Contact mode	[54]
	2205 HT	1 M NaCl solution	Contact mode	[55]
	Cutting edges of adhesively bonded Zn (Z) and Zn–Al–Mg (ZM) galvanized steel substrates	0.001 M NaCl solution	Tapping mode	[56]
	Ti_6_Al_4_V_2_Nd	1.5 wt% NaCl solution	Tapping mode	[57]
	EN AW-3003 and Al–Mn–Si–Zr fin alloy	SWAAT solution of pH 4	Contact mode	[58]

**Table 5 materials-13-00668-t005:** Summary of the application of EC-AFM in intercrystalline corrosion.

	Materials	Solution	Mode	Reference
Intercrystalline corrosion	AISI 304 stainless steel	1.0 wt% NaCl solution		[59]
	304L stainless steel	0.6 M and 1 M HNO_3_		[60]
	High-chromium cast iron	3.0 wt% NaCl solution	Contact mode	[61]
	Al_x_CoCrFeNi HEAs	3.5 wt% NaCl solution	Tapping mode	[63]
	CoCrMo alloys	Phosphate-buffered saline solution	Contact mode	[64]

**Table 6 materials-13-00668-t006:** Summary of the application of EC-AFM in coating protection.

	Materials	Coating	Solution	Mode	Reference
Coating protection	Carbon steel	Solvent-borne alkyd composite coating containing 1.0 wt% CeO_2_ nanoparticles (NPs) and 1.0 wt% polyaniline (PANI)	3.0 wt% NaCl solution	Contact mode	[66]
	Carbon steel	1.0 wt% *p*-toluene sulfonic acid (PTSA)-doped PANI in solvent-borne alkyd composite coating	3.0 wt% NaCl solution	Contact mode	[67]
	Carbon steel	Waterborne acrylic composite coating with 1 wt% acetic acid-stabilized CeO_2_ nanoparticles	3.0 wt% NaCl solution	Contact mode	[68]
	Austenitic stainless steel	Nanocrystalline (NC) and polycrystalline (PC) microstructure coatings	3.5 wt% NaCl solution	Tapping mode	[69]
	304L stainless steel	Magnetron-sputtered NC	0.05 M H_2_SO_4_ + 0.2 M NaCl solution	Tapping mode	[72]

**Table 7 materials-13-00668-t007:** Summary of the application of EC-AFM in corrosion inhibitor protection.

	Materials	Corrosion Inhibitor	Solution	Mode	Reference
Corrosion inhibitor protection	Copper	Linear sodium heptanoate	0.1 M Na_2_SO_4_ solution	Contact Mode	[74]
	Pure cast aluminum	Polyacrylic acid (PAA) + KI	0.5 M H_2_SO_4_	Tapping Mode	[76]
	AISI321 stainless steel	36% ethylene glycol–water solution	Ethylene glycol–water solution	Tapping Mode	[77]
	Copper	Ethyl-2-cyano-3-(4-(dimethylamino) phenyl) acrylate (ECDPA) and ZnO nanosheet composites	1 M HCl		[78]
	Copper	Dibenzylsulfoxide (DBSO) and *p*-chlorobenzohydroxamic acid (*p*-Cl-BHA)	0.5 M NaCl and 0.1 M Na_2_SO_4_	Contact Mode	[79]
	Polycrystalline copper	Benzotriazole (BTA)	0.5 M H_2_SO_4_	Contact Mode	[80]
	Copper	Benzotriazole (BTA)	0.01 M NaHCO_3_	Contact Mode	[81]

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
