# Peer review of "Application of Electrochemical Atomic Force Microscopy (EC-AFM) in the Corrosion Study of Metallic Materials"

_materials, 2020, doi:10.3390/ma13030668_

Round 1
Reviewer 1 Report
Remarks:
Authors are requested to read carefully the manuscript and correct all the language issues (ask the native speaker if necessary). A number of phrases broken in the middle can be found (i.e. page 15: “In the study of Davoodi and co-workers. The application of the EC-AFM and SECM integrated system for in-situ studies of the influence of intermetallic particles on local corrosion of aluminum alloys is introduced [46].”, “In the study of Shi and co-workers. Localized corrosion of AlxCoCrFeNi HEAs was studied by in-situ EC-AFM in 3.5wt % NaCl solution.”).
Missing spaces can be also found. Some extra commas would make phrases more clear. Sometimes words in the middle of the phrase begin with capital with no reason: “In addition, Based on the reasonable model that the surface potential under tensile stress would be higher than that under compressive stress.”, “However, At the sharp edge, electrons flow into a larger space in the matrix, corresponding to a larger total electron flow, which greatly increases the corrosion rate of the sharp edge.”
Also some phrases begin with small letters: “the uneven surface structure at the micro-scale, namely the lath-like structure, provides a favorable place for the ZnO/Zn(OH)2 formation and pitting.”
Are the formulas 1-7 developed by authors and presented for the first time in this paper? As this is very unlikely, the source should be provided. Also as EC-AFM is introduced in page 2, appropriate reference should be provided. Authors mention also other AFM techniques, such as EFM, MFM KPFM and so one. But the description suggests, that those techniques derive from EC-AFM. As this is not the case, appropriate relation in whole systematic should be presented. Authors mention that the EC-AFM can be utilized in so called constant height mode. Despite the fact that this mode was developed and in some early works presented in application, no practical aspects could be listed in order to justify its utilization in EC-AFM, as too many disadvantages causing artifacts appearance , lack of quantitative structures height and finally likely tip’s wear and damage could occur. More details about the way the EC-AFM works and what parameters and in what range can be controlled in the majority of commercial systems, should be provided. Authors use TappingMode name in inappropriate way and in general description of the tip-sample force detection method. As TappingMode is a trademark and the patent is owned by Bruker, it would be justified to use more common term such as intermittent tapping or semicontact mode. Figure 11 should have vertical (height, potential) scale. Units are written in inappropriate way: mv instead of mV, v instead of V as well as names: volta instead Volta. Main author of citation 47 is not appropriate. Also reference list should be verified in order to correct all typing issues.
Author Response
Authors are requested to read carefully the manuscript and correct all the language issues (ask the native speaker if necessary). A number of phrases broken in the middle can be found (i.e. page 15: “In the study of Davoodi and co-workers. The application of the EC-AFM and SECM integrated system for in-situ studies of the influence of intermetallic particles on local corrosion of aluminum alloys is introduced [46].”, “In the study of Shi and co-workers. Localized corrosion of AlxCoCrFeNi HEAs was studied by in-situ EC-AFM in 3.5wt % NaCl solution.”).
Missing spaces can be also found. Some extra commas would make phrases more clear. Sometimes words in the middle of the phrase begin with capital with no reason: “In addition, Based on the reasonable model that the surface potential under tensile stress would be higher than that under compressive stress.”, “However, At the sharp edge, electrons flow into a larger space in the matrix, corresponding to a larger total electron flow, which greatly increases the corrosion rate of the sharp edge.”
Also some phrases begin with small letters: “the uneven surface structure at the micro-scale, namely the lath-like structure, provides a favorable place for the ZnO/Zn(OH)2 formation and pitting.”
Response: As for some typos and format errors proposed, we have made corresponding modifications and highlighted all changes to the text with yellow (black text on yellow) in the revised manuscript.
Are the formulas 1-7 developed by authors and presented for the first time in this paper? As this is very unlikely, the source should be provided. Also as EC-AFM is introduced in page 2, appropriate reference should be provided.
Response: According to your suggestion, we have added relevant reference appropriately, as follows.
[2] Moore, S.; Burrows, R.; Picco, L.; Martin, T. L.; Greenwell, S. J.; Scott, T. B.; Payton, O. A study of dynamic nanoscale corrosion initiation events using HS-AFM. Faraday discussions 2018, 210, 409-428.
[11] Utsunomiya, T.; Yokota, Y.; Fukui, K.-i. Electrochemical Atomic Force Microscopy. In Compendium of Surface and Interface Analysis, 2018; pp 73-78.
Authors mention also other AFM techniques, such as EFM, MFM KPFM and so one. But the description suggests, that those techniques derive from EC-AFM. As this is not the case, appropriate relation in whole systematic should be presented.
Response: We changed that these technologies were developed from AFM, “On the basis of different capabilities, alarge amount of techniques has been developed after the invention of AFM, such as Kelvin Probe Force Microscopy (KPFM), Magnetic Force Microscopy (MFM), Scanning Electrochemical Microscopy-Atomic Force Microscopy (SECM-AFM), Electrostatic Force Microscopy (EFM), EC-AFM. These technologies are not only imaging tools, but also can accurately and quantitatively measure local physical and chemical phenomena.”
Authors mention that the EC-AFM can be utilized in so called constant height mode. Despite the fact that this mode was developed and in some early works presented in application, no practical aspects could be listed in order to justify its utilization in EC-AFM, as too many disadvantages causing artifacts appearance , lack of quantitative structures height and finally likely tip’s wear and damage could occur. More details about the way the EC-AFM works and what parameters and in what range can be controlled in the majority of commercial systems, should be provided.
As the reviewer said, there are very few practical applications of the constant height model in the literature we have compiled. However, as a model of early development, it has been mentioned in many literatures, thus it should be involved to ensure the integrity of this review. We made a table to summarize the operation mode of EC-AFM and added the necessary parameters. In the field of corrosion, there are mainly related with contact mode and intermittent tapping (tapping mode).
Table 1. Comparison of common operation modes of EC-AFM
|
Mode |
Probes (nominal spring constants) |
Cantilever |
Tip–sample distance |
Force |
scan rate |
|
Contact |
Silicon nitride on-contact mode) |
The cantilever is |
< 0.5 nm |
Repulsive |
High scan speeds |
|
Non-contact |
Silicon probes |
The cantilever is |
1–10 nm |
Attractive |
Slower scan speed than |
|
Iintermittent tapping(tapping |
Silicon probes |
The cantilever is |
0.5–2 nm(intermittent |
Both repulsive |
Slower scan speed |
Authors use TappingMode name in inappropriate way and in general description of the tip-sample force detection method. As TappingMode is a trademark and the patent is owned by Bruker, it would be justified to use more common term such as intermittent tapping or semicontact mode.
Response: According to your suggestion, TappingMode has been replaced with intermittent tapping.
Figure 11 should have vertical (height, potential) scale. Units are written in inappropriate way: mv instead of mV, v instead of V as well as names: volta instead Volta. Main author of citation 47 is not appropriate. Also reference list should be verified in order to correct all typing issues.
Response: There were no vertical (height, potential) scales in the original literature, mainly because the author wanted to exhibit that the pits in Figure (c) could correspond to the white points in Figure (a). We have corrected the unit errors as you mentioned, as well as the main author of citation 47 (now citation 50). According to your suggestion, we rechecked the reference list and corrected typing issues.
Reviewer 2 Report
In my opinion, authors should correct the following issues:
General remarks
#1) The paper is quite long. In my opinion, authors should summarize the paper, showing the main achievements of the selected papers, paying special attention to the discussion of obtained results and highlighting the advantages of using a EC-AFM in such studies.
#2) Authors should discuss properly all the results shown in the manuscript. Many figures extracted from references are not discussed in the manuscript, for instance, Fig. 3e, 4a, 4b,….14d, 19d, 20d.
Minor changes:
#1) please add a blank space before defining an acronym, for instance “atomic force microscope (AFM)” instead of “atomic force microscope (AFM)” [2nd paragraph, page 2]. Please revise all the manuscript accordingly.
#2) Please use always the same size for words throughout the paper. In the manuscript, authors use different size for some words (see, for instance, the end of 1st paragraph of section 2) or reference numbering in the main text (…[1]…). Please revise the paper accordingly.
#3) Please, add a blank space before STM. “that requires the samples to be conductive.STM can only directly observe”.
#4) For the sake of clarity, Please, define in the main text all the acronyms appearing in Figure 1. The definition of “CE” and “RE” is missed.
#5) Please, give numerical values for “but the thickness of the inner layer becomes thicker, leading to an increase in the thickness of the oxide film” [last paragraph, page 5]
#6) Please use subindex for “H2O2” [last paragraph, page 6]. Please revise the manuscript accordingly, for instance, Figure 24 caption.
#7) There is a typo in “and the corrosion depth of each phase is suppressed, as shown in Figure 10.” [1st paragraph, page 11]. I guess authors refers to Figure 7.
#8) Please use capital letter for “Intergranular” in “within 0.5 seconds during a galvanostatic scan. intergranular pits are distributed” [2nd paragraph, section 3.1.4, page 23]
#9) Please remove the dot after co-workers and the capital letter in “In the study of Shi and co-workers. Localized corrosion of AlxCoCrFeNi HEAs” [1st paragraph, page 25]
#10) Please use a capital letter at the beginning of the 1st paragraph of section 3.2.2.
#11) Please delete the sentence “We hope that these brief introductions will be helpful to future researchers and their work.” in the conclusion section.
Author Response
General remarks
#1) The paper is quite long. In my opinion, authors should summarize the paper, showing the main achievements of the selected papers, paying special attention to the discussion of obtained results and highlighting the advantages of using a EC-AFM in such studies.
Response: We added a summary at the end of each section to highlight the advantages of EC-AFM in this section as follows. In addition, we have listed summary tables.
3.1.1 EC-AFM has great advantages in studying the formation process and microstructure of corrosion product film on metal surface, especially in characterizing the evolution of early corrosion product film, which is conducive to analyzing the corrosion behavior of metal and promoting the research on corrosion resistance of metal. The application of EC-AFM in corrosion product film is summarized in Table 1.
3.1.2 EC-AFM can provide detailed information of the localized dissolution associated with different kinds of intermetallic particles, and deposition of corrosion products surrounding large particles or covering small pits, including the location of pitting initiation at controlled potentials, and whether pits are randomly distributed at nanoscale. The application of EC-AFM in pitting corrosion is summarized in Table 2.
3.1.3 The combination of EC-AFM and scanning Kelvin probe force microscopy (SKPFM) can simultaneously obtain topographical changes and Volta potential maps, which helps to better understand selective corrosion behavior and its mechanism. The application of EC-AFM in selective corrosion is summarized in Table 3.
3.1.4 EC-AFM can clearly reveal the formation of trenches and the local dissolution of the grain/phase interface, and explore the causes and mechanisms of intercrystalline corrosion in combination with other test tools. Table 4 summarizes the application of EC-AFM in intercrystalline corrosion.
3.2.1 EC-AFM can not only detect the state of the coating surface by high-resolution imaging, but also produce the coating defects by means of probe scraping to obtain the direct information of the coating corrosion resistance. The application of EC-AFM in coating protection is summarized in Table 5.
3.2.2 Through the morphology test of EC-AFM, the adsorption-desorption state of corrosion inhibitor on the electrode surface can be intuitively understood, which is very beneficial to the exploration of electrochemical mechanism of corrosion inhibitor. The application of EC-AFM in corrosion inhibitor protection is summarized in Table 6.
#2) Authors should discuss properly all the results shown in the manuscript. Many figures extracted from references are not discussed in the manuscript, for instance, Fig. 3e, 4a, 4b,….14d, 19d, 20d.
Response: We adjusted and summarized the content of the figures appropriately, and simply expressed what the author wanted to express, as follows:
The profile lines measured over the corroding intermetallic particle is shown in Figure 3e, which is helpful to observe the formation and roughness changes of the corrosion product film.
Moreover, amorphous alloys only have a few deep pits and no shallow pits as shown in Figure 4a and Figure 4c, while crystalline alloys have many deep pits and shallow pits, as shown in Figure 4b and Figure 4d.
cross-sectional profiles of pits along a–b lines in Figure 14a and c–f lines in Figure 14c are shown in Figure 14d. Since no carbide particles were found in the pit according to the cross-sectional profiles, they might have been dissolved with pit growth.
The protruded large I remained to be stable during the anodization, whereas the small II in Figure 19a were dissolved upon the anodization. Moreover, upon the anodization at 1 V, some active dissolution started at certain sites (e.g. III) forming small holes shown in Figure 19b. During the anodization at 2, 4 and 8 V (Figure 19c–19e), the small hole in area III became deeper as displayed by the profile lines in Figure 19f, and pronounced localized dissolution occurred in area IV. two small particles remained stable in the dissolved area (area IV in Figure 19d–19e). Besides, in the area marked as V, localized dissolution at one site resulted in a small hole, and a deposited particle of a few μm in size formed (Figure 19c), but it disappeared at 4V, exposing a much deeper pit (Figure 19d).
Figure 20d shows the depth line profile from the image obtained after the polarization at 1.2 VAg/AgCl, presenting a depth of ca. 200 nm between dissolved ferrite phase and the remaining austenite phase.
Minor changes:
#1) please add a blank space before defining an acronym, for instance “atomic force microscope (AFM)” instead of “atomic force microscope (AFM)” [2nd paragraph, page 2]. Please revise all the manuscript accordingly.
#2) Please use always the same size for words throughout the paper. In the manuscript, authors use different size for some words (see, for instance, the end of 1st paragraph of section 2) or reference numbering in the main text (…[1]…). Please revise the paper accordingly.
#3) Please, add a blank space before STM. “that requires the samples to be conductive.STM can only directly observe”.
#4) For the sake of clarity, Please, define in the main text all the acronyms appearing in Figure 1. The definition of “CE” and “RE” is missed.
#6) Please use subindex for “H2O2” [last paragraph, page 6]. Please revise the manuscript accordingly, for instance, Figure 24 caption.
#7) There is a typo in “and the corrosion depth of each phase is suppressed, as shown in Figure 10.” [1st paragraph, page 11]. I guess authors refers to Figure 7.
#8) Please use capital letter for “Intergranular” in “within 0.5 seconds during a galvanostatic scan. intergranular pits are distributed” [2nd paragraph, section 3.1.4, page 23]
#9) Please remove the dot after co-workers and the capital letter in “In the study of Shi and co-workers. Localized corrosion of AlxCoCrFeNi HEAs” [1st paragraph, page 25]
#10) Please use a capital letter at the beginning of the 1st paragraph of section 3.2.2.
#11) Please delete the sentence “We hope that these brief introductions will be helpful to future researchers and their work.” in the conclusion section.
Response: As for the minor changes proposed, we have made corresponding modifications and highlighted all changes to the text with yellow (black text on yellow) in the revised manuscript.
#5) Please, give numerical values for “but the thickness of the inner layer becomes thicker, leading to an increase in the thickness of the oxide film” [last paragraph, page 5]
Response: We have added numerical value “When the film-forming potential changes from -0.1 V(SCE) to 0.5 V(SCE), the chemical composition does not change, but the thickness of the inner layer becomes thicker, leading to an increase in the thickness of the oxide film from about 5 nm to 5.8 nm, and the compactness is improved, which makes the film more protective. ”
Reviewer 3 Report
A thorough revision of the state-of-the-art and recent breakthroughs on electrochemical-atomic force microscopy (EC-AFM) applied to the characterization of the solid-liquid interface in metal oxides is presented in the manuscript by Chen et al. The authors exhibit a remarkable dominion on the literature including some own relevant publications related to the topic. The subsequent qualitative and quantitative interpretation of the results reported herein is accurate and reliable. I do not have any serious criticisms regarding the summary of the key results, methodologies, conclusions, references, clarity or context. The experimental procedures and short but precise discussions described in the article comprise a good showcase of the technique when applied to corrosion processes. The separation of the manuscript in different sub-sections, namely corrosion product film, pitting corrosion, selective corrosion, coating protection and so on, is consistent, meaningful, and orientative. The paper is of enough interest and puts together valuable original works that merit its publication in Materials. Nevertheless, prior to be considered for its publication, some minor revisions of the current version of the manuscript should be addressed by the authors. The latter can be summarized in the following bullet points:
Although it is clear that this review is mostly oriented to specialists, some sentences exposing the particularities of the counter cathodic reactions (equations 3 to 7 in page 2) would be desired It would be interesting to address certain questions related to the design of the electrochemical cell for the EC-AFM setup. Specifically, those related to the disposal, placement, and geometries of the counter electrode in order to ensure a proper and homogeneous distribution of the EM field lines… Some typos such as: (i) It is well known that AFM is a kind of scanning probe microscope(SPM) and as an extension of the scanning tunneling microscope(STM) developed by Binnig, Quate, and Gerber in 1986 (page 2). (ii) and there is a similar increase in surface roughness after the application of external potential, which is mainly due to the formation of oxyhydroxide layer (page 6). (iii) In the study of Liu and the co-workers (page 6). (iv) The formation of the corrosion product film and tends to be uniform (page 11). (v) The topography and surface roughness of DSS 2205 and AISI 316L changed with the increase of chloride-ion concentration were examined (page 11) and so on… The discussion of reference 35 (page 7) is a bit unclear and needs to be rewritten The contribution by Izquierdo et al. (reference 39) is undeniable relevant. However, it makes reference to a related but different technique, i.e. AFM in combination with scanning electrochemical microscopy (SECM). Consequently, it falls a bit out of the scope of this review… No significant differences can be deduced form the descriptions made by the authors of the studies included in references 47 and 48… In some parts of the text the authors claim a real time characterization of the topography of the sample under an applied potential. However, taking into account the time scale associated with the collection of (conventional)high-resolution AFM images (they normally require a lapse time of several minutes), a call of attention for the readers on the prospective applications of “real” high-speed AFM (time range of seconds or milliseconds) to polarized solid-liquid interfaces could be of interest. The latter is timidly mentioned on page 23 when the authors allude to reference 56, but it should be highlighted noticeably… Characteristic limitations associated to the technique should be extended to more than a little paragraph in the conclusions in order to show the readers a whole brochure of the possibilities of the EC-AFM technique as complete/ realistic as possible
Author Response
A thorough revision of the state-of-the-art and recent breakthroughs on electrochemical-atomic force microscopy (EC-AFM) applied to the characterization of the solid-liquid interface in metal oxides is presented in the manuscript by Chen et al. The authors exhibit a remarkable dominion on the literature including some own relevant publications related to the topic. The subsequent qualitative and quantitative interpretation of the results reported herein is accurate and reliable. I do not have any serious criticisms regarding the summary of the key results, methodologies, conclusions, references, clarity or context. The experimental procedures and short but precise discussions described in the article comprise a good showcase of the technique when applied to corrosion processes. The separation of the manuscript in different sub-sections, namely corrosion product film, pitting corrosion, selective corrosion, coating protection and so on, is consistent, meaningful, and orientative. The paper is of enough interest and puts together valuable original works that merit its publication in Materials. Nevertheless, prior to be considered for its publication, some minor revisions of the current version of the manuscript should be addressed by the authors. The latter can be summarized in the following bullet points:
Although it is clear that this review is mostly oriented to specialists, some sentences exposing the particularities of the counter cathodic reactions (equations 3 to 7 in page 2) would be desired
Response: According to your suggestion, we have added some content to describe the redox reactions involved in the equation, as follows:The basic process of metallic corrosion in an aqueous solution consists of the anodic dissolution of metals and the cathodic reduction of oxidants. The redox reactions (equations. 1-7) involve the transfer of electrons and ions between the metal and the solution. According to the corrosion kinetics, the anodic oxidation current of the metal degradation is equal to the cathode reduction current of the oxidant at the corrosion potential. When the metal electrode potential is more positive, the rates of cathodic reactions increase and the rates of anodic reactions decrease accordingly. Conversely, as the metal electrode potential becomes more negative, the effect on the reactions is opposite.
It would be interesting to address certain questions related to the design of the electrochemical cell for the EC-AFM setup. Specifically, those related to the disposal, placement, and geometries of the counter electrode in order to ensure a proper and homogeneous distribution of the EM field lines…
Response: According to your suggestion, we have added the content about the electrode as follows: Generally, the electrochemical cell is made from chemical-resistant polycarbonate, which can be used with a wide variety of liquids. Eight-degree nose assemblies are recommended for imaging in liquid because the smaller angle considers the different angle the laser makes as it goes in and out of the fluid, compared to operation in air. The electrochemical cell typically contains retaining clips, O-ring gasket and the liquid cell plate. When assembled, the sample itself comprises the bottom of the liquid container. Therefore, the sample must be large enough for the O-ring to seat. The work electrode should be relatively small, each point on the work electrode should be geometrically equivalent to the counter electrode, which ensures that the current and potential across the electrode are evenly distributed. Others are based on conventional electrochemical testing requirements.
Some typos such as: (i) It is well known that AFM is a kind of scanning probe microscope(SPM) and as an extension of the scanning tunneling microscope(STM) developed by Binnig, Quate, and Gerber in 1986 (page 2). (ii) and there is a similar increase in surface roughness after the application of external potential, which is mainly due to the formation of oxyhydroxide layer (page 6). (iii) In the study of Liu and the co-workers (page 6). (iv) The formation of the corrosion product film and tends to be uniform (page 11). (v) The topography and surface roughness of DSS 2205 and AISI 316L changed with the increase of chloride-ion concentration were examined (page 11) and so on…
Response: As for the Some typos proposed, we have made corresponding modifications and highlighted all changes to the text with yellow (black text on yellow) in the revised manuscript.
The discussion of reference 35 (page 7) is a bit unclear and needs to be rewritten
Response: We have rewritten reference 35 (now ref 38) as follows: In addition, Bearinger et al. [38] characterized hydration of titanium/titanium oxide surfaces under freely corroding and potentiostatically held conditions using EC-AFM. In contrast to conventional high vacuum techniques, EC-AFM enables measurement of morphological surface structure in the in situ hydrated state. The results show that the titanium surface covers the oxidized dome and grows laterally during hydration. Applied potential altered the growth rate. Under open circuit potential conditions, growth proceeded approximately six times faster than under a -1 V applied voltage. The oxide growth is partly due to the lateral expansion and overgrowth of the dome at the interface of oxide-solution. The method was successfully used to study dynamic changes in the surface morphology.
The contribution by Izquierdo et al. (reference 39) is undeniable relevant. However, it makes reference to a related but different technique, i.e. AFM in combination with scanning electrochemical microscopy (SECM). Consequently, it falls a bit out of the scope of this review…
Response: AFM-SECM is regarded as a particularly attractive tool in corrosion science, as the advantages of both techniques, the unambiguous topographic information provided by AFM and the laterally resolved electrochemical information obtained by SECM. The integrated system used EC-AFM to obtain the change of sample morphology. The change of morphology is obtained when applied voltage is provided by potentiostat. As mentioned in ref 42 and ref 39(now ref 43): “we present the potential of AFM-SECM for studying corrosion processes induced by anodically activated surfaces. ” and “The use of the external CH832A bipotentiostat imposed that the copper surface could not be maintained under potentiostatic control at all times, since the cell had to be intermittently switched off in order to change the potential of the working electrodes. ”
No significant differences can be deduced form the descriptions made by the authors of the studies included in references 47 and 48…
Response: According to your suggestion, we re-examined the two papers, and there were indeed significant similarities, so we deleted one to make sure there were no duplicates.
In some parts of the text the authors claim a real time characterization of the topography of the sample under an applied potential. However, taking into account the time scale associated with the collection of (conventional)high-resolution AFM images (they normally require a lapse time of several minutes), a call of attention for the readers on the prospective applications of “real” high-speed AFM (time range of seconds or milliseconds) to polarized solid-liquid interfaces could be of interest. The latter is timidly mentioned on page 23 when the authors allude to reference 56, but it should be highlighted noticeably… Characteristic limitations associated to the technique should be extended to more than a little paragraph in the conclusions in order to show the readers a whole brochure of the possibilities of the EC-AFM technique as complete/ realistic as possible
Response: According to your suggestion, we have added some introduction about HS-AFM in page 26: Based on EC-AFM, the contact mode high speed-AFM (HS-AFM) compensates for the shortcomings of AFM with short collection times. The long collection time is a limiting factor for AFM. Contact-mode HS-AFM images multiple frames per second, which is orders of magnitude faster than traditional AFM. This enables real-time imaging processes with nano-scale lateral resolution and sub-nanometer-scale height resolution. The increase in speed can not only directly image dynamic nanoscale events, but also macroscopic regions of the sample surface without reducing resolution. It is a valuable imaging tool for in-situ observation of nanoscale corrosion initiation events such as metastable pitting, grain boundary dissolution, and short crack formation during stress corrosion cracking.
Reviewer 4 Report
This review focuses on direct observation studies of corrosion behaviors at metallic surfaces by electrochemical atomic force microscopy (EC-AFM), which provides useful information about material science. This review may be published after considering the following points.
The “Application of EC-AFM” section describes each work study-by-study, which makes it difficult for the readers to understand the general insights. The observed materials, solution, type and mode of AFM used (in-situ or ex-situ; contact or tapping), etc. should be summarized in a table. The authors should make the list of abbreviations in the manuscript because there are many mistakes. Some abbreviations and parameters are used without definition: ‘SCE’ in Page 4, ‘OCP’ in Page 9, parameter ‘i’ in the caption of Fig. 12 (Page 14), ‘SVET’ and ‘EIS’ in Page 22, and ‘FCC’ and ‘BCC’ in Page 25, ‘Vsat’ in Page 27, and ‘FTIR’ in Page 38. Furthermore, the definitions of some abbreviations are duplicated: ‘X-ray photoelectron spectroscopy (XPS)’ in Pages 6 and 33, ‘simulated inflammation (SI)’ in Page 6, ‘nanocrystalline (NC)’ in Pages 31-32 (four times!), ‘coarse crystal (CC)’ in Page 32, and ‘benzotriazole (BTA)’ in Pages 37-38. Although ‘CeO2 nanoparticles (NPs)’ is described in Page 29, this term is re-defined as ‘CeNP’ in Page 30. Defining this as ‘CeNPs’ may be better. The definition of ‘stainless steel (SS)’ was described in Page 32, but ‘SS’ was already used in Page 6. In Page 4, regarding non-contact-mode AFM, the authors state “The force between the tip and the sample is Van der Waals attraction. In the non-contact mode, it has no damage to the surface of the sample and the lateral force is the smallest, but the resolution is low and the scanning speed is slow.” However, recent development and improvement of the equipment enable us to detect repulsive forces, to obtain atomic-resolution images, and to conduct high-speed observation of sample surfaces in solution by non-contact-mode AFM [for example, ACS Nano 12, 11785 (2018)]. With this method, for example, dissolution processes of calcite in water were traced with atomic resolution [Nano Lett. 17, 4083 (2017)], which indicates that this method is potentially applicable to the observation of corrosion processes at metal surfaces. This may be described somewhere (probably in the perspective section). The authors should carefully recheck the manuscript because there are many grammatical and format errors. The examples are as follows: Use appropriate subscripts and superscripts for ‘Al88Ni8Ce4’ in Page 9 (caption of Fig. 5), ‘Cu2+’ in Page 10, ‘A/m2’ in Page 14, ‘ClO4-’ in Page 16, and much more. Use appropriate capitals for ‘In-situ’ and ‘volta’ in Page 17 and ‘intergranular’ in Page 23, ‘corrosion inhibitor is…’ in Page 33. Some sentences are accidentally ended after ‘coworkers’ in Pages 15, 32, and 34. Equations 1 and 7; ‘n’ of ‘ne’ should be displayed in italic. Equations 2 and 7; use periods instead of commas. Page 2; the phrases `developed by Binnig, …` and ‘invented by Binnig, …’ are duplicated in the same paragraph. The former can be deleted. Page 2; using ‘and so on’ is inappropriate. Page 4; ‘corrosion product film’ and ‘corrosion products film’ should be ‘a/the corrosion product film’ or ‘corrosion product films’. Page 5, Line 2 from the bottom; ‘0.1 V(SCE)’ may be ‘-0.1 V(SCE)’? Page 6; ‘Liu and coworkers’ should be ‘Liu and Gilbert’. Page 7; ‘Bearinger and co-worker’ should be ‘Bearinger et al.’. P16; The authors should refer to Figure 16 elsewhere in the main text. P20, caption of Fig. 20; ‘Troom’ should be ‘room temperature’. In the reference list, there are many omissions (journal names, pages, and years).
Author Response
This review focuses on direct observation studies of corrosion behaviors at metallic surfaces by electrochemical atomic force microscopy (EC-AFM), which provides useful information about material science. This review may be published after considering the following points.
The “Application of EC-AFM” section describes each work study-by-study, which makes it difficult for the readers to understand the general insights. The observed materials, solution, type and mode of AFM used (in-situ or ex-situ; contact or tapping), etc. should be summarized in a table. The authors should make the list of abbreviations in the manuscript because there are many mistakes.
Response: According to your suggestion, we have summarized your questions and added tables to the article, as shown in the following tables. The applications of EC-AFM in the field of corrosion we mentioned are in situ studies. Besides, we have also corrected the abbreviations.
Table 2. Summary of the application of EC-AFM in corrosion products film
|
|
Materials |
Solution |
Mode |
Ref |
|
Corrosion product film |
carbon steel |
carbonate/bicarbonate solution |
contact mode |
Ref 33 |
|
304L stainless steel |
nitric acid |
|
Ref 34 |
|
|
AA 2024-T3 |
0.5 M NaCl solution |
tapping mode |
Ref 35 |
|
|
CoCrMo alloy |
phosphate buffer saline (PBS) and simulated inflammation (SI) solution |
contact mode |
Ref 36 |
|
|
Ti-6Al-4V |
PBS |
contact mode |
Ref 37 |
|
|
Ni50Nb50 metallic glass |
1 mol/L HCl solution |
tapping mode |
Ref 39 |
|
|
Al–Ni–Ce metallic glass |
0.01 mol/L NaCl solution |
tapping mode |
Ref 40 |
|
|
X100 pipeline steel |
0.01 M NaHCO3 solution |
contact mode |
Ref 41 |
|
|
NAB |
3.5 wt. % NaCl solution |
contact mode |
Ref 44 |
Table 3. Summary of the application of EC-AFM in pitting corrosion
|
|
Materials |
Solution |
Mode |
Ref |
|
Pitting corrosion |
AISI 316L and DSS 2205 |
physiological solution and artificial saliva |
contact mode |
Ref 45 |
|
304L stainless steel |
chloride borate buffer solution |
contact mode |
Ref 46 |
|
|
SUS304 |
3.5 wt. % NaCl solution |
|
Ref 48 |
|
|
pure aluminum |
0.01 mol/L FeC13 solution |
contact mode |
Ref 49 |
|
|
AA1050 |
10 mM NaCl+ 5 mM KI |
contact mode |
Ref 50 |
|
|
Metallic zinc |
0.5 M NaOH solution containing different concentrations of ClO3- or ClO4- anions |
|
Ref 51 |
|
|
hot-dip-galvanized surface |
0.01 M NaOH |
contact mode |
Ref 52 |
Table 4. Summary of the application of EC-AFM in selective corrosion
|
|
Materials |
Solution |
Mode |
Ref |
|
Selective corrosion |
AZ91 alloy added (La,Ce) mischmetal |
0.1 mol/L NaCl solution |
contact mode |
Ref 53 |
|
Al 6060 alloy |
0.2 M Na2SO4 solution |
contact mode |
Ref 54 |
|
|
2205 HT |
1 M NaCl solution |
contact mode |
Ref 55 |
|
|
cutting-edges of adhesively bonded Zn (Z) and Zn-Al-Mg (ZM) galvanized steel substrates |
0.001 M NaCl solution |
tapping mode |
Ref 56 |
|
|
Ti6Al4V2Nd |
1.5 wt. % NaCl solution |
tapping mode |
Ref 57 |
|
|
EN AW-3003 and Al–Mn–Si–Zr fin alloy |
SWAAT solution of pH 4 |
contact mode |
Ref 58 |
Table 5. Summary of the application of EC-AFM in intercrystalline corrosion
|
|
Materials |
Solution |
Mode |
Ref |
|
Intercrystalline corrosion |
AISI 304 stainless steel |
1.0 wt. % NaCl solution |
|
Ref 59 |
|
304L stainless steel |
0.6 M and 1 M HNO3 |
|
Ref 60 |
|
|
high chromium cast iron |
3.0 % NaCl solution |
contact mode |
Ref 61 |
|
|
AlxCoCrFeNi HEAs |
3.5 wt. % NaCl solution |
tapping mode |
Ref 63 |
|
|
CoCrMo alloys |
phosphate-buffered saline solution |
contact mode |
Ref 64 |
Table 6. Summary of the application of EC-AFM in coating protection
|
|
Materials |
coating |
Solution |
Mode |
Ref |
|
Coating protection |
carbon steel |
solvent-borne alkyd composite coating containing 1.0 wt.% CeO2 nanoparticles (NPs) and 1.0 wt.% polyaniline (PANI) |
3.0 wt. % NaCl solution |
contact mode |
Ref 66 |
|
carbon steel |
1.0 wt. % p-toluene sulfonic acid (PTSA) doped PANI in solvent-borne alkyd composite coating |
3.0 wt. % NaCl solution |
contact mode |
Ref 67 |
|
|
carbon steel |
waterborne acrylic composite coating with1.0 wt. % acetic acid-stabilized CeO2 nanoparticles |
3.0 wt. % NaCl solution |
contact mode |
Ref 68 |
|
|
austenitic stainless steel |
nanocrystalline (NC) and polycrystalline (PC) microstructures coating |
3.5 wt. % NaCl solution |
tapping mode |
Ref 69 |
|
|
304 stainless steel |
magnetron sputtered NC |
0.05 M H2SO4 + 0.2 M NaCl solution |
tapping mode |
Ref 72 |
Table 7. Summary of the application of EC-AFM in corrosion inhibitor protection
|
|
Materials |
Corrosion inhibitor |
Solution |
Mode |
Ref |
|
Corrosion inhibitor protection |
copper |
linear sodium heptanoate |
0.1 M Na2SO4 solution |
contact mode |
Ref 74 |
|
pure cast aluminum |
polyacrylic acid (PAA) + KI |
0.5 M H2SO4 |
tapping mode |
Ref 76 |
|
|
AISI321 stainless steel |
36% ethylene glycol-water solution |
ethylene |
tapping mode |
Ref 77 |
|
|
copper |
Ethyl-2-cyano-3-(4-(dimethylamino) phenyl) acrylate(ECDPA)and ZnO nanosheet composites |
1 M HCl |
|
Ref 78 |
|
|
copper |
dibenzylsulphoxide (DBSO) and p-chlorobenzohydroxamic acid (p-Cl-BHA) |
0.5 M NaCl and 0.1 M Na2SO4 |
contact mode |
Ref 79 |
|
|
polycrystalline copper |
benzotriazole (BTA) |
0.5 M H2SO4 |
contact mode |
Ref 80 |
|
|
copper |
benzotriazole (BTA) |
0.01 M NaHCO3 |
contact mode |
Ref 81 |
Some abbreviations and parameters are used without definition: ‘SCE’ in Page 4, ‘OCP’ in Page 9, parameter ‘i’ in the caption of Fig. 12 (Page 14), ‘SVET’ and ‘EIS’ in Page 22, and ‘FCC’ and ‘BCC’ in Page 25, ‘Vsat’ in Page 27, and ‘FTIR’ in Page 38. Furthermore, the definitions of some abbreviations are duplicated: ‘X-ray photoelectron spectroscopy (XPS)’ in Pages 6 and 33, ‘simulated inflammation (SI)’ in Page 6, ‘nanocrystalline (NC)’ in Pages 31-32 (four times!), ‘coarse crystal (CC)’ in Page 32, and ‘benzotriazole (BTA)’ in Pages 37-38. Although ‘CeO2 nanoparticles (NPs)’ is described in Page 29, this term is re-defined as ‘CeNP’ in Page 30. Defining this as ‘CeNPs’ may be better. The definition of ‘stainless steel (SS)’ was described in Page 32, but ‘SS’ was already used in Page 6.
Response: We have modified the abbreviations and defined some parameters.
In Page 4, regarding non-contact-mode AFM, the authors state “The force between the tip and the sample is Van der Waals attraction. In the non-contact mode, it has no damage to the surface of the sample and the lateral force is the smallest, but the resolution is low and the scanning speed is slow.” However, recent development and improvement of the equipment enable us to detect repulsive forces, to obtain atomic-resolution images, and to conduct high-speed observation of sample surfaces in solution by non-contact-mode AFM [for example, ACS Nano 12, 11785 (2018)]. With this method, for example, dissolution processes of calcite in water were traced with atomic resolution [Nano Lett. 17, 4083 (2017)], which indicates that this method is potentially applicable to the observation of corrosion processes at metal surfaces. This may be described somewhere (probably in the perspective section).
Response: According to your suggestion, we have added the relevant content in page 4 and emphasize that this method may be applicable to the observation of metal corrosion process: However, the development and improvement of this device in recent years have enabled non-contact mode to detect repulsion and obtain atomic resolution images. Miyata used a non-contact-mode AFM to observe the dissolution of calcite in water at high speed. This may well be applicable to the observation of corrosion processes on metal surfaces.
The authors should carefully recheck the manuscript because there are many grammatical and format errors. The examples are as follows: Use appropriate subscripts and superscripts for ‘Al88Ni8Ce4’ in Page 9 (caption of Fig. 5), ‘Cu2+’ in Page 10, ‘A/m2’ in Page 14, ‘ClO4-’ in Page 16, and much more. Use appropriate capitals for ‘In-situ’ and ‘volta’ in Page 17 and ‘intergranular’ in Page 23, ‘corrosion inhibitor is…’ in Page 33. Some sentences are accidentally ended after ‘coworkers’ in Pages 15, 32, and 34. Equations 1 and 7; ‘n’ of ‘ne’ should be displayed in italic. Equations 2 and 7; use periods instead of commas. Page 2; the phrases `developed by Binnig, …` and ‘invented by Binnig, …’ are duplicated in the same paragraph. The former can be deleted. Page 2; using ‘and so on’ is inappropriate. Page 4; ‘corrosion product film’ and ‘corrosion products film’ should be ‘a/the corrosion product film’ or ‘corrosion product films’. Page 5, Line 2 from the bottom; ‘0.1 V(SCE)’ may be ‘-0.1 V(SCE)’? Page 6; ‘Liu and coworkers’ should be ‘Liu and Gilbert’. Page 7; ‘Bearinger and co-worker’ should be ‘Bearinger et al.’. P16; The authors should refer to Figure 16 elsewhere in the main text. P20, caption of Fig. 20; ‘Troom’ should be ‘room temperature’. In the reference list, there are many omissions (journal names, pages, and years).
Response: As for the grammatical and format errors proposed, we have made corresponding modifications and highlighted all changes to the text with yellow (black text on yellow) in the revised manuscript. According to your suggestion, we rechecked the reference list and corrected typing issues.
Round 2
